

# Total column water vapor retrieval for GOME-2 visible blue observations

Ka Lok Chan[a], Pieter Valks[a], Sander Slijkhuis[a], Claas Köhler[a], and Diego Loyola[a]

[a]Remote Sensing Technology Institute, German Aerospace Center (DLR), Oberpfaffenhofen, Germany

**Correspondence:** Ka Lok Chan (ka.chan@dlr.de)

**Abstract.**

We present a new total column water vapor (TCWV) retrieval algorithm in the visible blue spectral band for the Global Ozone Monitoring Experience 2 (GOME-2) instruments on board the EUMETSAT MetOp satellites. The blue band algorithm allows retrieval of water vapor from sensors which do not cover longer wavelengths, such as Ozone Monitoring Instrument (OMI) and the Copernicus atmospheric composition missions Sentinel-5 Precursor (S5P), Sentinel-4 (S4) and Sentinel-5 (S5). The blue band algorithm uses the differential optical absorption spectroscopic (DOAS) technique to retrieve water vapor slant columns. The measured water vapor slant columns are converted to vertical column using air mass factors (AMFs). The new algorithm has an iterative optimization module to dynamically find the optimal a priori water vapor profile. This makes it better suited for climate studies than usual satellite retrievals with static a priori or vertical profile information from chemistry transport model (CTM). The dynamic a priori algorithm makes use of the fact that the vertical distribution of water vapor is strongly correlated to the total column. The new algorithm is applied to GOME-2A and GOME-2B observations to retrieve TCWV. The data set is validated by comparing to the operational product retrieved in the red spectral band, sun-photometer and radiosonde measurements. Water vapor columns retrieved in the blue band are in good agreement with the other data sets, indicating that the new algorithm derives precise results, and can be used for the current and forthcoming Copernicus Sentinel missions S4 and S5.

## 1 Introduction

Atmospheric water vapor is the most important natural greenhouse gas in the troposphere, accounting for more than 60 % of the greenhouse effect (Clough and Iacono, 1995; Kiehl and Trenberth, 1997). Despite this importance, its roles in climate and its reactions to climate change are still difficult to assess. As the atmosphere gets warmer, water vapor contents are expected to rise faster than the total precipitation amount, which is governed by the surface heat budget through evaporation (Trenberth and Stepaniak, 2003). This results in a 'positive water vapor feedback' that further amplify the original warming effect (Colman, 2003; Soden et al., 2005; Soden and Held, 2006). On the other hand, clouds are known to have positive effects on cooling the earth's surface (Bellomo et al., 2014; Brown et al., 2016). However, the net cooling or warming effect of clouds in a continuous warming atmosphere is not yet well understood (Boucher et al., 2013). To investigate these complex interactions and evaluate



climate models, continuous monitoring of the spatio-temporal variations of total column water vapor (TCWV) on a global scale is necessary (Hartmann et al., 2013).

Satellite remote sensing observations are an effective way to monitor the spatio-temporal variations of column amount water vapor on a global scale. High quality water vapor data can be derived from a large number of satellite sensors operating in various wavelength regions (optical, infrared, microwave) (Kaufman and Gao, 1992; Bauer and Schluessel, 1993; Noël et al., 1999, 2004; Li et al., 2006; Wagner et al., 2006; Pougatchev et al., 2009; Wang et al., 2014; Grossi et al., 2015). Each sensor has its specific advantages and limitations, be it for spatio-temporal resolution, for truly global coverage, for sensitivity, or for the long timelines required for climate monitoring. An extensive overview of satellite measurement of water vapor can be found in Schröder et al. (2018).

In this work, we focus on the development of water vapor retrieval algorithm for spectroscopic satellite observations in the ultraviolet (UV) and visible (Vis) spectral range with nadir viewing geometry. This kind of observation has long been conducted since the Global Ozone Monitoring Experience (GOME) mission launched in 1995 (Burrows et al., 1999). Together with other follow up satellite missions, for example, SCanning Imaging Absorption SpectroMeter for Atmospheric CHartographY (SCIAMACHY) (Bovensmann et al., 1999), Global Ozone Monitoring Experience 2 (GOME-2) (Callies et al., 2000) and Ozone Monitoring Instrument (OMI) (Levelt et al., 2006), these observations provide a global record of earthshine radiance in the UV and Vis spectral range for more than 25 years. The recent satellite mission TROPOspheric Monitoring Instrument (TROPOMI) (Veefkind et al., 2012) on board the European Space Agency (ESA) Sentinel 5 Precursor (S5P) satellite provides daily global observations of earthshine radiance in the UV and Vis range with much finer spatial resolution ($3.5\,\mathrm{km} \times 7\,\mathrm{km}$) compared to its predecessors. The TROPOMI/S5P and the upcoming Sentinel 5 (S5) missions will provide indispensable global observations of earthshine radiance in the UV and Vis ranges in the next decade. Retrieving TCWV from these observations can provide important independent data sets for climate studies and contribute towards TCWV climate data records (Beirle et al., 2018; Schröder et al., 2018).

TCWV is typically retrieved in the visible red and near infrared (NIR) spectral range (Grossi et al., 2015). As most of the current and forthcoming sensors do not cover the red band, it is necessary to develop a new water vapor retrieval in the available spectral bands. Most of the spectroscopic satellite borne instruments, e.g., GOME, GOME-2, OMI, TROPOMI, etc, cover the blue spectral band as it is essential for the monitoring of major atmospheric pollutants, i.e., nitrogen dioxide ($NO_2$) (Richter and Burrows, 2002; Valks et al., 2011; Boersma et al., 2011; Krotkov et al., 2017). Retrieving TCWV in this wavelength band can provide a consistent long time series of climate record from similar type of satellite sensors. Figure 1 shows the water vapor absorption cross section in the UV and Vis bands together with the spectral band available to the current GOME-2, OMI and S5P sensors as well as the forthcoming S4 and S5 instruments. The red shadowed area indicates the spectral range used in the current GOME-2 operational water vapor retrieval. The blue shadowed area denotes the wavelength band used to retrieve TCWV in this study. Previous studies have demonstrated the feasibility of retrieving water vapor slant columns and total columns from GOME-2 and OMI satellite observations in the blue band (Wagner et al., 2013). Based on the similar approach, Wang et al. (2014) has derived TCWV from OMI observations using a priori information from the Goddard Earth Observing



System version 5 (GEOS-5) model assimilation product. Details of the spectral analysis settings and retrieval parameters used in previous studies and this work are shown in Table 1.

**Figure 1.** Water vapor absorption cross section. (a) The horizontal bars show the spectral band available to varies satellite sensors. The wavelength range used in this study and the operational GOME-2 product are highlighted in blue and red color, respectively. Zoom in of (a) to the blue and red bands are shown in (b) and (c), respectively. The red curves show the water vapor absorption cross section convoluted with the instrument slit function. Note that the scale of the y axis of each plot is different.



**Table 1.** Parameters and settings of water retrieval in the blue band used in previous studies and this work.

| Instrument | Spectral range | Parameter | A priori profile | reference |
|---|---|---|---|---|
| GOME-2 & OMI | 430 - 450 nm | Slant column only | N.A. | Wagner et al. (2013) |
| OMI | 430 - 480 nm | Slant & total column | GEOS-5 model | Wang et al. (2014) |
| OMI | 427.7 - 465 nm | Slant & total column | MERRA-2 model | Wang et al. (2016) |
| GOME-2 | 427.7 - 455 nm | Slant & total column | Statistical analysis | This work |

The objective of this study is to develop a TCWV retrieval algorithm for spectroscopic satellite observations which fulfills the following requirements. Firstly, the algorithm should be feasible for the current and forthcoming satellite sensors, such as OMI, S5P, S4 and S5. Secondly, the retrieval should not rely on input from chemistry transport model (CTM) to avoid propagating model error into the climatological measurement records. Lastly, the retrieval should provide a realistic error estimation as

measurement uncertainty is an important parameter for data assimilation and future harmonization of satellite data. Based on the results from previous studies, we have further optimized spectral analysis settings for the TCWV retrieval and developed a statistical analysis approach to optimize the a priori water vapor profile used in the retrieval. In addition, a comprehensive error estimation is also included in the new water vapor retrieval algorithm. The developed algorithm has been implemented to retrieve TCWV from GOME-2 observations; in the future we will extend the application to other similar satellite sensors. For

validation, the new TCWV data set retrieved from GOME-2 observations are compared to the GOME-2 operational product, ground based sun-photometer and radiosonde measurements.

The paper is organized as follows. Section 2 describes all instruments and data sets used in this study. The concept of the TCWV retrieval is presented in Section 3.1. The description of the spectral retrieval of water vapor slant columns is shown in Section 3.1.1. Section 3.1.2 presents the iterative optimization method for the conversion of satellite measurement

of water vapor slant columns to total columns. A detailed error estimation is presented in Section 3.1.7. The validation of the GOME-2 TCWV is shown in Section 4. Section 4.1 presents the comparison against the GOME-2 operational product. The comparison against sun-photometer and radiosonde data are shown in Section 4.2 and Section 4.3, respectively. Discussions of the discrepancies between different data sets are presented in Section 5. Finally, the conclusion is drawn in Section 5.3.

## 2   Instruments and data sets

### 2.1   The GOME-2 instruments

The Global Ozone Monitoring Experiment 2 (GOME-2) are passive nadir viewing satellite borne spectrometers on board the European Organization for the Exploitation of Meteorological Satellites (EUMETSAT) MetOp series of satellites. The MetOp satellites orbit at an altitude of ~820 km on sun-synchronous orbits with 29 days (412 orbits) repeat cycle and a local equator overpass time of 09:30 LT (local time) on the descending node. MetOp-A, the first MetOp satellite, was launched on

$19^{th}$ October 2006. MetOp-B was launched 6 years later on $17^{th}$ September 2012. The third MetOp satellite, MetOp-C, was





launched on $7^{th}$ November 2018. All GOME-2 instruments are currently in operation. A more detailed introduction of the MetOp series of satellites can be found in Klaes et al. (2007).

The GOME-2 instruments are optical spectrometers equipped with scanning mirrors which enable across-track scanning in nadir and side ways viewing for polar coverage (Callies et al., 2000). Each GOME-2 instrument consists of four detectors

covering a wavelength range of 240 - 790 nm with spectral resolution ranging from 0.26 nm to 0.51 nm. The nominal spatial resolution of the instruments is 80 km (across-track) × 40 km (alongtrack) for the forward scan and the spatial resolution reduced to 240 km (across-track) × 40 km (alongtrack) for the backward scan. The scanning swath width of the GOME-2 instruments is about 1920 km. After the GOME-2 instrument on board the MetOp-B satellite (refers as GOME-2B from hereafter) went in tandem operation with MetOp-A in July 2013, the across-track spatial resolution of the GOME-2 instrument on board the

MetOp-A satellite (refers as GOME-2A from hereafter) was doubled with the spatial coverage of a swath reduced to 960 km. The spatial resolution and coverage of GOME-2B remains unchanged. A more detailed description of the GOME-2 instruments can be found in Munro et al. (2016). In this study, we focus on the results from GOME-2A as it provides longer term observations. GOME-2B results are shown mainly for the investigation of the consistency between the sensors.

## 2.2 GOME-2 level 1B data

The first step of GOME-2 data processing is the conversion of detector signal (level 0 data) to geolocation and radiometric calibrated radiance and irradiance data (level 1B data). GOME-2 observations taken before $25^{th}$ June 2015 were processed by the level 1B processor version 6.0, while GOME-2 data taken after $25^{th}$ June 2015 were processed by the updated level 1B processor version 6.1. The processor update mainly resolved spectral artefacts in the GOME-2 on-ground calibration key data. The effect of the spectral contamination in level 1B data processed by version 6.0 processor is significant at the blue band

(Band 3) (Azam et al., 2015). The improvement of level 1B data has been reported to have a significant impact on the $NO_2$ retrieval in the blue band, reducing the $NO_2$ columns by 6 - 23 % (Liu et al., 2019).

## 2.3 Operational GOME-2 TCWV product

The operational GOME-2 water vapor product is processed by the German Aerospace Center (DLR) within the framework of EUMETSAT's Satellite Application Facility on Atmospheric Composition Monitoring (AC-SAF), using the GOME Data

Processor (GDP) version 4.8. The product is used as reference to validate the TCWV retrieved in the blue band. The operational algorithm retrieves water vapor slant columns in the wavelength range of 614 - 683 nm. The conversion of slant columns to vertical columns uses air mass factors (AMFs) derived from oxygen slant columns measured in the same spectral band. Water vapor absorption in the red band is much stronger (more than an order of magnitude) than that in the blue spectral range (see Figure 1), thus, yielding better signal to noise ratios. In addition, the retrieval of water vapor in the red band uses air mass

factors derived from oxygen measurements at the same wavelength range which reduces the dependency on the numerical calculation of radiative transfer in the atmosphere (Grossi et al., 2015). The operational GOME-2 water vapor product has been validated intensively by radiosonde and Global Positioning System (GPS) measurements (Antón et al., 2015; Román et al., 2015; Kalakoski et al., 2016; Vaquero-Martínez et al., 2018). The operational product has been reported to significantly





underestimate the TCWV over central Africa and India; it overestimates the TCWV over oceans in the tropics during summer of the northern hemisphere (Grossi et al., 2015). Compared to radiosonde and GPS data, the operational GOME-2 water vapor product has in general a dry bias of 3 - 11 % (Antón et al., 2015; Román et al., 2015; Kalakoski et al., 2016; Vaquero-Martínez et al., 2018).

## 2.4   Sun-photometer measurements

The CIMEL CE-318 sun-photometers are used in the AERosol RObotic NETwork (AERONET) to measure direct sun and sky radiance at multiple wavelengths (Holben et al., 1998). These sun-photometer observations do not only provide information of aerosol optical properties (Holben et al., 2001) but also of columnar water vapor content (Alexandrov et al., 2009). Water vapor columns are retrieved from sun-photometer observations in the near infrared (NIR) at 940 nm where water vapor absorption is rather strong. The inversion of water vapor columns is based on the attenuation of radiation through the atmosphere. A more detailed description of the water vapor retrieval algorithm can be found in Alexandrov et al. (2009). Water vapor columns are provided in the standard AERONET product. The AERONET water vapor product has also been validated by microwave radiometry, GPS and radiosondes measurements (Pérez-Ramírez et al., 2014). The sun-photometer measurements are in general underestimating the columnar water vapor by 6 - 9 % (Pérez-Ramírez et al., 2014). Cloud screened and quality assured level 2.0 data are used in this study. In this work, all AERONET stations providing co-located columnar water vapor measurements from 2008 to 2018 are used to validate the new GOME-2 water vapor retrieval results. In total, there are 905 AERONET stations providing co-located data with GOME-2. The locations of these AERONET stations are indicated in Figure 2 as red triangles.

## 2.5   Radiosonde measurements

Radiosonde data are taken from the Integrated Global Radiosonde Archive version 2 (IGRA2) database. The database is managed by the National Centers for Environmental Information (NCEI) of the National Oceanic and Atmospheric Administration (NOAA). The IGRA2 database includes quality assured radiosonde measurements from over 2700 globally distributed stations. The measurements consist of temperature, relative humidity, dew point depression, wind direction and wind speed at multiple pressure levels. The IGRA2 radiosonde data are publicly available on the webpage of NCEI (https://www.ncdc.noaa.gov/data-access/weather-balloon/integrated-global-radiosonde-archive). A more detailed description of the radiosonde data can be found in Durre et al. (2006). Compared to ground based observations, the radiosonde measurements of TCWV show an error of ~5 % with bias ranging from -1.19 kg/m$^2$ to 1.01 kg/m$^2$ (Wang and Zhang, 2008; Van Malderen et al., 2014). In this study, all radiosonde stations providing co-located columnar water vapor measurements from 2008 to 2018 are used to validate the GOME-2 water vapor measurements in the blue band. The locations the 578 radiosonde stations providing co-located data are indicated in Figure 2 as blue circles.





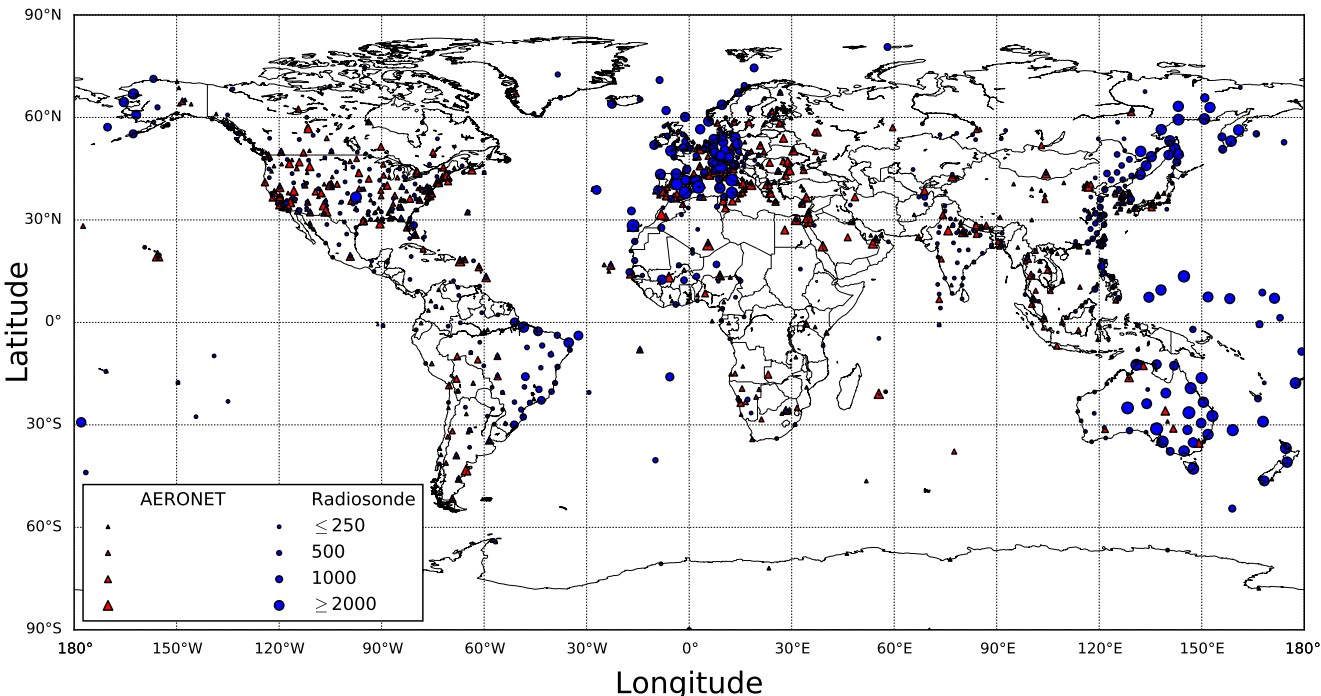

**Figure 2.** Locations of sun-photometer (red triangles) and radiosonde (blue circles) stations providing co-located TCWV measurements with GOME-2 satellite observations. The size of the markers is proportional to the number of valid observations available.

## 2.6 ERA-Interim reanalysis data

ERA-Interim is a global atmospheric reanalysis data set produced by the European Centre for Medium-Range Weather Forecasts (ECMWF) (Dee et al., 2011; Berrisford et al., 2011). The ERA-Interim reanalysis data covers a long time period since 1979, providing consistent data on a global scale for the analysis of long term variation of water vapor in the atmosphere. The

5    reanalysis data is produced with a data assimilation scheme which combined varies measurements as prior information from model forecasts. The original data set is in a spatial resolution of $\sim 80\,\mathrm{km}$ (T255 Spectral). The data is then transformed to the latitude longitude (LL) coordinate system with a horizontal resolution of $0.75° \times 0.75°$ through the ECMWF's Meteorological Archival and Retrieval System (MARS). TCWV is retrieved from the system with a temporal resolution of 6 hours. The ERA-Interim data from 2008 to 2018 is used in the statistical analysis of water vapor vertical distribution and the relation to their

10    total column amount.





## 3  Methodology

### 3.1  The blue band TCWV retrieval

The GOME-2 water vapor retrieval algorithm in the blue spectral range follows the classical differential optical absorption spectroscopy (DOAS) approach, which is a standard spectroscopic method for the retrieval of weakly absorbing trace gases (Platt and Stutz, 2008). The method consists of two major steps. The first step is the retrieval of water vapor slant columns. The second step is the conversion of the water vapor slant columns to vertical columns. A comprehensive error estimation is also included in the retrieval. Details of the retrieval algorithm and error estimation are presented in the following.

### 3.1.1  Water vapor slant column retrieval

Typical absorption spectroscopy describes the attenuation properties of radiation along an optical path by the Beer-Lambert-Bouguer law. For satellite measurements, the equation can be written as Equation 1.

$$I(\lambda) = I_0(\lambda) \cdot exp\left(-\varepsilon_M(\lambda) - \varepsilon_R(\lambda) - L\sum_{i=1}^{n}\sigma_i(\lambda)c_i\right) \cdot R(\lambda) \tag{1}$$

$I_0(\lambda)$ refers to the direct sun irradiance spectrum taken at the top of atmosphere (TOA), while $I(\lambda)$ is the earthshine radiance spectrum taken by looking down from space towards the nadir direction measuring sun light reflected by the earth's surface and atmosphere. $L$ represents the effective optical path length from TOA to the earth's surface and reflected from the earth's surface back to the satellite. $\sigma_i$ denotes the absorption cross section of gas $i$, and $c_i$ is its average concentration along the effective optical path. $\varepsilon_M$ and $\varepsilon_R$ are the Mie and Rayleigh extinction integrated along the light path, respectively. $R(\lambda)$ represents the reflectance of the earth. The optical density $\tau(\lambda)$ can then be calculated by taking logarithm of the ratio between $I_0(\lambda)$ and $I(\lambda)$ as shown in Equation 2.

$$\tau(\lambda) = ln\left(\frac{I_0(\lambda) \cdot R(\lambda)}{I(\lambda)}\right) \tag{2}$$

In practice Equation 1 cannot be directly applied for trace gas retrieval, as some of the extinction processes, i.e., Mie and Rayleigh scattering, are not quantified. The DOAS method unitizes the fact that atmospheric scattering processes only show broad band spectral characteristics while trace gases exhibit narrow band absorption structures (Platt and Stutz, 2008). Therefore, the optical density $\tau(\lambda)$ can be separated into narrow (or differential) $\tau'(\lambda)$ and broad $\tau_b(\lambda)$ band contributions. The broad band contribution $\tau_b$ can be approximated by a low order polynomial $p(\lambda)$. The broad band structures in $R(\lambda)$ can also be accommodated by $p(\lambda)$ and narrow band features in $R(\lambda)$ can be included as pseudo cross sections in the spectral fit. Thus, the equation can be rewrite as Equation 3.

$$\tau(\lambda) = \tau'(\lambda) + \tau_b(\lambda) = L \cdot \sum_{i=1}^{n}\sigma_i(\lambda)c_i + p(\lambda) \tag{3}$$





Characteristic absorption features of different trace gases are then used to determine their concentrations $c_i$ along the effective optical path $L$.

Slant column densities (SCDs) of water vapor are retrieved from GOME-2 spectra by applying the DOAS spectral fitting technique. The SCD is defined as the integrated concentration along the optical path from TOA through the atmosphere to the earth's surface and reflected back to the satellite sensor ($L \times c_i$). The DOAS spectral fit is applied to the wavelength range of 427.7 - 455 nm. The following absorption cross sections are employed in the DOAS fit: water vapor at 293 K from HITEMP database (Rothman et al., 2010) and scaled by Lampel et al. (2015), $NO_2$ at 220 K (Vandaele et al., 2002), $O_3$ at 228 K (Brion et al., 1998), $O_4$ at 293 K (Thalman and Volkamer, 2013), and liquid water at 297 K (Pope and Fry, 1997) as well as a Ring spectrum. Two additional GOME-2 polarization key data are also included in the DOAS fit to correct for remaining level 1B calibration issues caused by polarization. These cross sections are first convoluted with the effective instrument slit function to the instrument spectral resolution. The effective slit function is derived by convolving a high-resolution reference solar spectrum (Chance and Kurucz, 2010) with a stretched preflight GOME-2 slit function and aligning to the GOME-2 daily irradiance measurements with stretch factors as fit parameters. Similar approaches with different spectral retrieval settings have also been used to retrieve slant column water vapor from different satellite sensors, e.g., GOME-2 and OMI (Wagner et al., 2013; Wang et al., 2014, 2016). A brief summary of the previous studies is presented in Table 1. An example of the spectral fitting retrieval of a GOME-2A spectrum taken on $1^{st}$ July 2008 over the Pacific Ocean is shown in Figure 3.

The spectral fitting window is optimized for water vapor retrieval which includes a relatively strong water vapor absorption structure at about 442 nm. Including liquid water absorption in the analysis effectively eliminates the interference of liquid water and reduces the systematic error above surfaces covered by water (Wang et al., 2014, 2016). The spectral fitting window is optimized to minimize the influence from spectral contamination in the GOME-2 level 1B data. This issue has been reported to be more significant for wavelengths longer than 460 nm (Azam et al., 2015) therefore the fitting window has been limited to 455 nm. A $4^{th}$ order polynomial is included in the DOAS fit to remove the broad band spectral structures of Rayleigh scattering and lower order Mie scattering, broad band trace gas absorption as well as instrumental effects. Using a polynomial with higher order is likely to improve the DOAS fit and minimize the fit residual, but it is difficult to justify the physical meaning. Shift and stretch parameters of radiance spectra are also fitted in the spectral fitting process to compensate for the instability due to small thermal variations of the spectrograph. The spectral fitting results are the slant columns of water vapor. Figure 4a shows the water vapor slant columns retrieved from GOME-2A observations on $1^{st}$ July 2008 (orbit 8813 - 8826). The corresponding slant column uncertainties and the root mean square of the spectral fit residual are shown in Figure 4b and c, respectively. As expected the retrieved water vapor slant columns show higher values over tropical regions and lower slant columns at upper latitudes. In addition, the slant column uncertainties and the root mean square of the spectral fit residual are significantly higher at both ends of the satellite orbits. There the observations are taken with very high solar zenith angle, and thus lower radiance intensity and signal to noise ratio. The mean spectral fitting uncertainty is about $5.2 \, kg/m^2$ over the tropics ($30°S$ - $30°N$) which is equivalent to mean relative error of $\sim$13.9 %. The average root mean square of the spectral fitting is $9.2 \times 10^{-4}$.

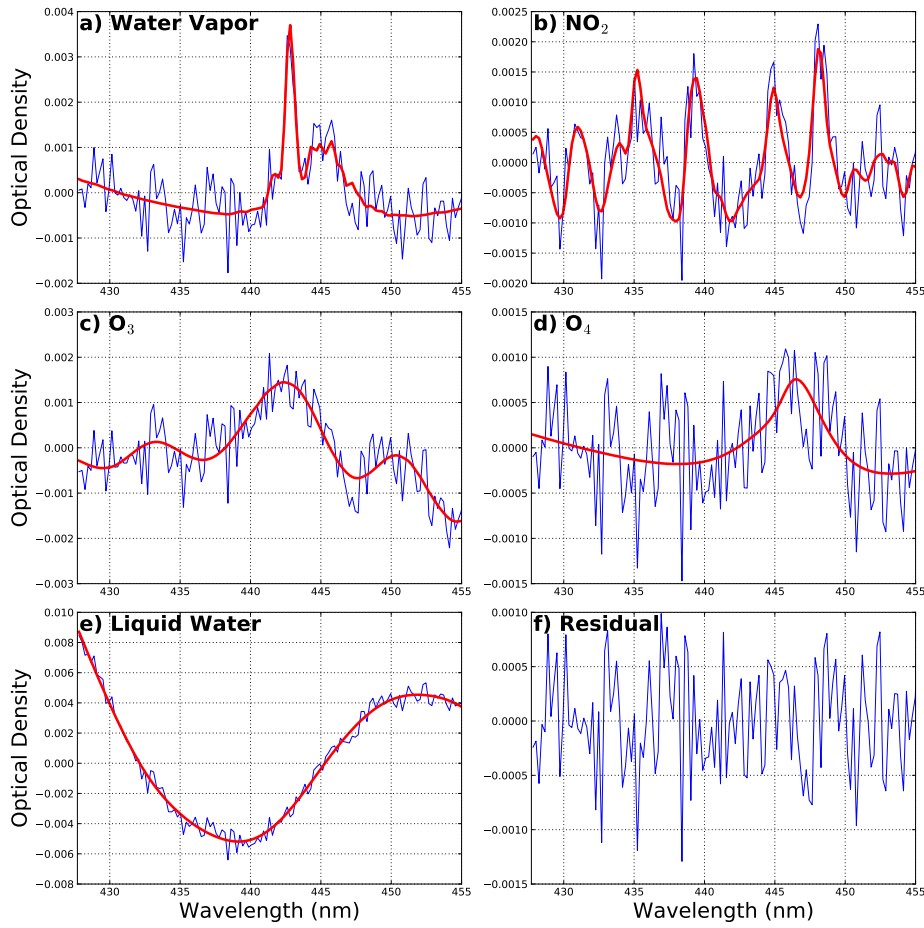

**Figure 3.** An example of the DOAS retrieval of water vapor slant column from a GOME-2A spectrum over the Pacific Ocean. The retrieved water vapor slant column is $30.6\,\mathrm{kg/m^2}$. The blue curves show measured optical density with broad band attenuation removed by subtracting a $4^{th}$ order polynomial and the red curves show the optical density of the scaled reference absorption cross sections.

### 3.1.2 Air mass factor

The next step of the TCWV retrieval is the conversion of water vapor SCDs to vertical column densities (VCDs). The VCD (or total column) is defined as the vertical integral of water vapor from the surface to the top of atmosphere. The SCD to VCD conversion is accomplished by using the concept of air mass factor (AMF) (Solomon et al., 1987). As water vapor SCDs

5 are retrieved within a relatively narrow spectral window, we can assume the wavelength dependency of the optical path is



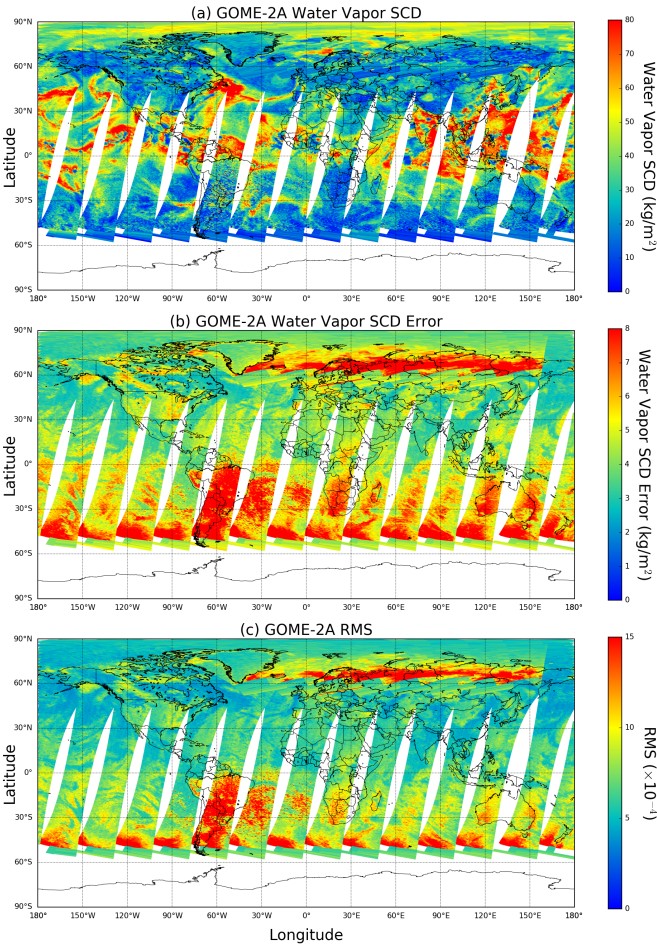

**Figure 4.** (a) Slant column densities of water vapor retrieved from GOME-2 observations on $1^{st}$ July 2008 (orbit 8813 - 8826). (b) The corresponding water vapor slant column uncertainties and (c) the root mean square of the spectral fit residual.

negligible. Thus, the AMFs need only be calculate at a representative wavelength. Due to the relatively strong water vapor absorption feature at 442 nm, the AMFs are calculated at this wavelength. The AMF can be expressed as Equation 4.

$$AMF = \frac{SCD}{VCD} \qquad (4)$$

Light traveling in the atmosphere can be scattered by air molecules, aerosols, and clouds, resulting in a complex optical path. To resolve the optical path as well as the box air mass factor ($\Delta$AMF), comprehensive multiple scattering radiative transfer calculations are required. The $\Delta$AMF is defined as the AMF of each individual vertical layer. Typically, the height dependent





air mass factor can be decoupled from the vertical distribution of optically thin absorbers (Palmer et al., 2001). As a result, the AMF can then be calculated from the $\Delta$AMF following Equation 5.

$$AMF = \frac{SCD}{VCD} = \frac{\sum\limits_{l=surface}^{l=TOA} \Delta AMF_l \times \Delta z_l \times c_l}{\sum\limits_{l=surface}^{l=TOA} \Delta z_l \times c_l} \tag{5}$$

where $\Delta z_l$ and $c_l$ are the thickness and the number density of the absorber at layer $l$, respectively. $c_l$ is taken from the a priori profile. The $\Delta$AMFs are independent of the vertical distribution of the absorber, but strongly dependent on viewing geometry, solar position, surface albedo and surface altitude.

### 3.1.3 Box air mass factor look-up table

The $\Delta$AMF can be calculated using a radiative transfer model. To reduce the processing time, $\Delta$AMFs are pre-calculated with a number of representative observation and solar geometries, surface albedo and surface pressure and stored in a look-up table. In the current version of retrieval algorithm, the $\Delta$AMF look-up table is calculated with the radiative transfer model VLIDORT version 2.7 (Spurr, 2008) at 442 nm with an aerosol free US standard atmosphere (Anderson et al., 1986). The $\Delta$AMFs for each particular GOME-2 observation can then be derived by interpolating within the look-up table. Details of the parameterization of the $\Delta$AMF look-up table are shown in Table 2.

For retrieval, the $\Delta$AMF look-up table is interpolated linearly in the surface albedo ($A_s$), relative azimuth angle ($\phi$), cosine of solar zenith angle ($\cos\theta$), and cosine of viewing zenith angle ($\cos\alpha$) dimensions, while a nearest neighbor interpolation is applied to the surface pressure dimension. In the current version of retrieval algorithm, surface albedo is taken from the climatology monthly minimum Lambertian equivalent reflector (LER) product version 2.1 at 440 nm derived from observations of the corresponding GOME-2 sensor (Tilstra et al., 2017) and spatially interpolated to the GOME-2 measurement locations. The GOME-2 surface LER (version 2.1) dataset takes the advantage of using more recent observations (2007 - 2013) and accounting the degradation of GOME-2 level 1 data. The GOME-2 surface LER (version 2.1) dataset is in a resolution of $0.5° \times 0.5°$ with an increased resolution of $0.25° \times 0.25°$ along coastlines. The viewing and solar geometries are taken from the GOME-2 level 1B product. The resulting $\Delta$AMF profile is then linearly interpolated to match the vertical grid of water vapor a priori profile. The AMF can then be calculated following Equation 5.

### 3.1.4 A priori water vapor vertical profile

The vertical distribution of water vapor is important for the conversion of slant columns water vapor to vertical columns as expressed in Equation 5. Most of the trace gas retrievals from satellite measurements in the UV and Vis spectral range





**Table 2.** Parameters in the box air mass factor look-up table.

| Parameter | Symbol | Number of grid points | Grid values |
|---|---|---|---|
| Viewing zenith angle (°) | $\alpha$ | 10 | 0, 10, 20, 30, 40, 50, 60, 65, 70, 75 |
| Solar zenith angle (°) | $\theta$ | 20 | 0, 10, 20, 30, 40, 45, 50, 55, 60, 65, 70, 72, 74, 76, 78, 80, 82, 84, 86, 88 |
| Relative azimuth angle (°) | $\phi$ | 7 | 0, 30, 60, 90, 120, 150, 180 |
| Surface albedo | $A_s$ | 14 | 0, 0.01, 0.025, 0.05, 0.075, 0.1, 0.15, 0.2, 0.25, 0.3 0.4, 0.6, 0.8, 1.0 |
| Surface pressure (hPa) | $P_s$ | 17 | 1063.10, 1037.90, 1013.30, 989.28, 965.83, 920.58, 876.98, 834.99, 795.01, 701.21, 616.60, 540.48, 411.05, 308.00, 226.99, 165.79, 121.11 |
| Pressure level (hPa) | $P_l$ | 64 | 1056.77, 1044.17, 1031.72, 1019.41, 1007.26, 995.25, 983.38, 971.66, 960.07, 948.62, 937.31, 926.14, 915.09, 904.18, 887.87, 866.35, 845.39, 824.87, 804.88, 785.15, 765.68, 746.70, 728.18, 710.12, 692.31, 674.73, 657.60, 640.90, 624.63, 608.58, 592.75, 577.34, 562.32, 547.70, 522.83, 488.67, 456.36, 425.80, 396.93, 369.66, 343.94, 319.68, 296.84, 275.34, 245.99, 210.49, 179.89, 153.74, 131.40, 104.80, 76.59, 55.98, 40.98, 30.08, 18.73, 8.86, 4.31, 2.18, 1.14, 0.51, 0.14, 0.03, 0.01, 0.001 |

use vertical profile information from chemistry transport model simulations (e.g., Wang et al., 2014, 2016; Krotkov et al., 2017; De Smedt et al., 2018). Previous studies uses a priori profile from GEOS-5 and MERRA-2 model products to retrieve TCWV from OMI observations (Wang et al., 2014, 2016). However, using a priori profile from model is not optimal for our water vapor retrieval, as it is supposed to be an independent 'climatological' product. We developed an iterative approach to

5  optimize the a priori water vapor vertical profile used in the satellite retrieval to make the satellite measurements independent from model simulations, and avoid propagating model errors into the measurement. The iterative a priori profile optimization approach is based on the statistical analysis of water vapor vertical distribution over 11 years from 2008 to 2018. Figure 5 shows the statistical analysis of water vapor profiles from ECMWF ERA-Interim reanalysis data (Dee et al., 2011; Berrisford et al., 2011) over a small region of Pacific Ocean (5°S - 5°N, 180°W - 170°W) in July of 2008 - 2018. Water vapor profiles are

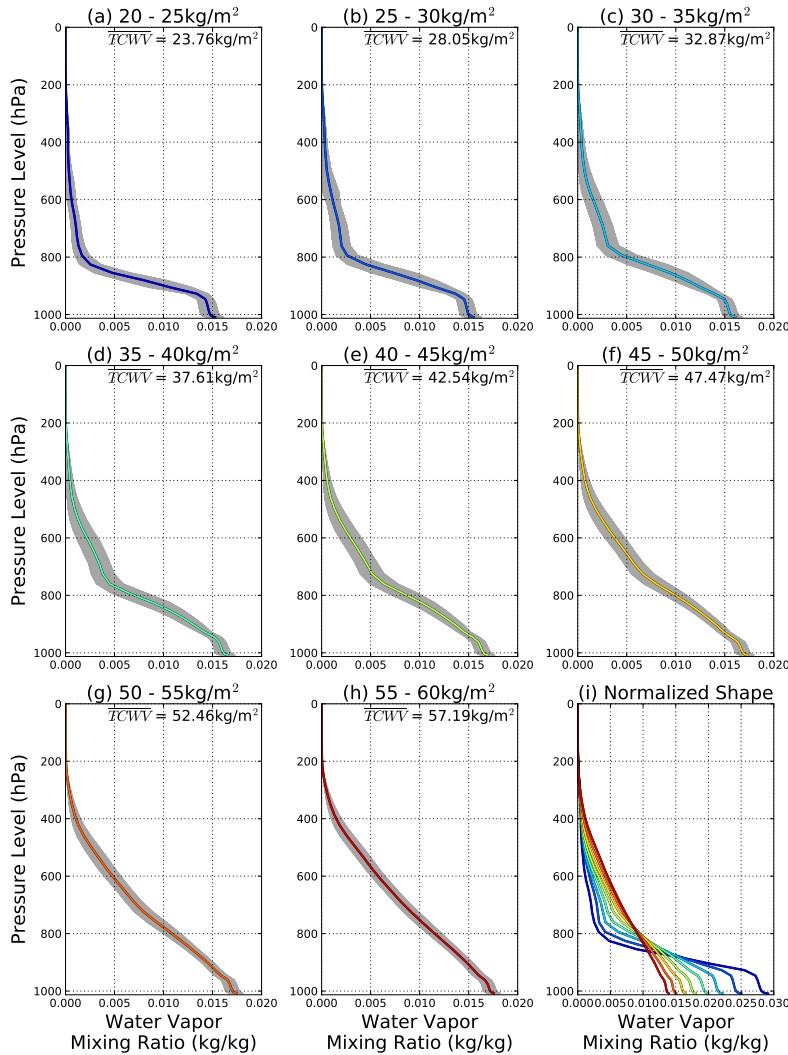

**Figure 5.** Statistical analysis of water vapor vertical profiles from ECMWF ERA-Interim reanalysis data over a small region of Pacific Ocean ($5°$S - $5°$N, $180°$W - $170°$W) in July of 2008 to 2018. Water vapor profiles are sorted by their total column density into 8 ranges, which are (a) 20 - 25 kg/m$^2$, (b) 25 - 30 kg/m$^2$, (c) 30 - 35 kg/m$^2$, (d) 35 - 40 kg/m$^2$, (e) 40 - 45 kg/m$^2$, (f) 45 - 50 kg/m$^2$, (g) 50 - 55 kg/m$^2$ and (h) 55 - 60 kg/m$^2$. Shadowed areas indicate the 1 $\sigma$ standard deviation variation of water vapor mixing ratio. (i) shows the normalized mean water vapor profile shapes of these 8 ranges.

sorted by their total column densities into 8 ranges from 20 kg/m$^2$ up to 60 kg/m$^2$. Color coded lines indicate the mean profile of each range, while shadowed areas represent the 1 $\sigma$ standard deviation variation of water vapor mixing ratio. The normalized mean profiles for each range are also indicated in Figure 5i. These profiles are normalized by dividing their total columns and multiplying with the mean total column calculated from all measurements. The analysis result shows that water vapor vertical



profile shapes are strongly related to their own column densities. Water vapor profiles with similar total columns show very similar vertical distribution. Water vapor is typically concentrated close to the surface below 800 hPa when the total column is small (i.e., less than 30 kg/m$^2$). It starts to extend to higher altitudes with increasing total column and changes the profile shape. A much larger portion of water vapor is located above 800 hPa when the total column is larger than 40 kg/m$^2$. The small
standard deviation of the water vapor mixing ratio profile also indicates that the water vapor profile shape only varies slightly within each range.

By making use of the characteristic that water vapor profile shapes are strongly correlated to their total columns, we have formulated a water vapor vertical profile shape look-up table for the entire globe with a spatial resolution of 0.75°. Water vapor profiles are sorted into five ranges for each geolocation and for each month of the year. The mean profiles, the total
columns, and the standard deviation of total columns for each range are stored in a look-up table. The water vapor vertical profile shape look-up table is interpolated linearly in the spatial dimension to the satellite measurement location for each range. The iterative optimization of a priori water profile begins by using the overall mean profile of the satellite measurement location of the corresponding month. This mean water vapor profile is then used together with the corresponding ΔAMFs to calculate an initial AMF following Equation 5. The water vapor slant column is divided by this initial AMF to retrieve the initial
vertical column. The look-up table is then linearly interpolated in the total column dimension to the retrieved initial column to retrieve the corresponding vertical profile shape. The interpolated profile is again used to retrieve the second vertical column. This process repeats until the difference between the input and output water vapor column is less than 1 % or the number of iteration reaches the limit. As the retrieval of more than 99 % of GOME-2 measurements stopped within 3 iterations, the limit of maximum number of iteration in the current version of retrieval is set to 5.

### 3.1.5   Partially cloudy scene observations

Clouds are treated as opaque Lambertian surfaces in the retrieval algorithm. The treatment of partially cloudy pixels is based on the independent pixel approximation (Martin et al., 2002; Boersma et al., 2004) where the pixel is separated into two independent parts, one with fully cloud cover and the other one is completely cloud free. Air mass factors are calculated
separately for both clear sky and cloudy parts. Cloud information, including cloud fraction ($CF$), cloud albedo ($A_c$), and cloud top pressure ($P_c$), are taken from the GOME-2 operational cloud product (Loyola et al., 2007, 2010; Lutz et al., 2016). The assumption of Lambertian cloud is more representative for optically thick clouds. Therefore, we transformed optically thin clouds to Lambertian equivalent clouds in the retrieval. As cloud albedo is directly related to the cloud optical thickness, cloud fractions are converted to effective cloud fraction ($CF_{eff}$) using the cloud albedo following Equation 6.

$$CF_{eff} = \frac{CF \times A_c}{0.8} \tag{6}$$

The cloudy AMF (AMF$_{cld}$) is calculated from the ΔAMF look-up table by setting the surface pressure to cloud top pressure and replacing the surface albedo with the cloud albedo. It should be noted that the same a priori water vapor profile is assumed





in both the cloudy AMF and clear sky AMF (AMF$_{clr}$) calculations. Following Equation 5, the calculation of SCD for the cloudy scene is insensitive to water vapor below cloud and ΔAMFs below cloud are 0. On the other hand, VCD is calculated by integrating the water vapor profile from the surface to the top of atmosphere which includes the part below cloud. This 'invisible' column below the cloud (also known as 'ghost column') is taken from the a priori profile.

AMFs of partially cloudy pixels are calculated as the intensity weighted average of the AMF$_{cld}$ and AMF$_{clr}$. This weighting is commonly know as intensity weighted cloud fraction ($CF_{iw}$) which is defined by Equation 7.

$$CF_{iw} = \frac{CF_{eff} \times I_{cld}}{CF_{eff} \times I_{cld} + (1 - CF_{eff}) \times I_{clr}} \tag{7}$$

where $I_{cld}$ and $I_{clr}$ represents the radiance intensity for the cloudy and clear sky scenes, respectively. The radiance intensities are pre-calculated using radiative transfer model VLIDORT at 442 nm for a number of representative observation and solar
geometries, surface albedo, and surface pressure and stored in a look-up table. The settings of the intensity look-up table are the same as the ΔAMF look-up table but without the pressure level dimension. The AMF can then be calculated following Equation 8.

$$AMF = AMF_{cld} \times CF_{iw} + AMF_{clr} \times (1 - CF_{iw}) \tag{8}$$

The resulting AMFs are used to divide the measured slant columns to convert water vapor slant columns to vertical columns.
This AMF is used for the iterative optimization of a priori profile of partially cloudy pixels.

### 3.1.6   Aerosol

The presence of aerosols affects the radiative transfer in the atmosphere and may influence the retrieval of surface properties, cloud and atmospheric water vapor (Bhatia et al., 2015; Bhatia et al., 2018). As the aerosol properties, e.g., extinction profile,
single scattering albedo, asymmetry parameter, etc., are unknown, there is no general and easy solution to explicitly account for aerosols in the retrieval. On the other hand, it is very difficult to separate cloud and aerosol in the cloud retrieval due to their similarity in optical properties. As a result, the aerosol effect is already implicitly considered in the cloud product (Boersma et al., 2004, 2011). Therefore, no additional treatment of aerosol is applied in the water vapor retrieval algorithm.

### 3.1.7   Error estimation

The error of the TCWV is composed of many sources. Major sources of error can be divided into two parts: one is related to the measurement itself and the other is related to the uncertainties of assumptions in the retrieval. The uncertainty of the TCWV can be derived analytically through error propagation. As the retrieval of TCWV is separated into two major steps,





slant column retrieval and AMF calculation, the error estimation also follows these two steps. The uncertainty of TCWV can be express as Equation 9.

$$\sigma_{vcd}^2 = VCD^2 \times \left( \left( \frac{\sigma_{scd}}{SCD} \right)^2 + \left( \frac{\sigma_{amf}}{AMF} \right)^2 \right) \tag{9}$$

where $\sigma_{vcd}$, $\sigma_{scd}$ and $\sigma_{amf}$ are the uncertainty of TCWV, the error of water vapor slant column, and air mass factor uncertainty,

respectively. Details of the estimation of the water vapor slant column uncertainty and air mass factor error are presented in the following.

### 3.1.8 Slant column error

The uncertainties of water vapor slant column are mainly attributed to the instrument noise, instrument characteristics, and the
uncertainties related to the DOAS retrieval of slant column. Instrument noise is expected to cause random error and this error can be quantified by analyzing the DOAS fit residual (Stutz and Platt, 1996). Other sources of error, related to the instrument, are the uncertainties of instrument slit function, incomplete removal of stray light, and wavelength calibration uncertainties. In addition, we have uncertainties of absorption cross sections and temperature dependency of the absorption cross sections. The contributions of systematic errors to the slant column uncertainties are estimated through sensitivity tests with absorption
cross section with different effective temperature and different assumptions of instrument slit function shape. We estimated the systematic error of the slant column is about 3 %. The total error of the slant column can be calculate following Equation 11.

$$\sigma_{scd}^2 = \sigma_{scd_r}^2 + (0.03 \times SCD)^2 \tag{10}$$

where $\sigma_{scd_r}$ is the random error estimated by analyzing the DOAS fit residual.

### 3.1.9 Clear sky air mass factor error

The uncertainty of the AMF is mainly related to the uncertainties of each input parameter used in the AMF calculation. These input parameters include the solar and viewing geometries, surface albedo, surface pressure, and water vapor vertical profile. The solar and viewing geometries are well calibrated and their errors are mainly related to the interpolation of the box AMF look-up table. These uncertainties are negligible compared to other sources of error. The contribution to the AMF uncertainty of
the remaining sources of error can be estimated by the AMF sensitivity (or Jacobian) with respect to each parameter (Boersma et al., 2004). The Jacobian is derived from the box air mass factor look-up table using the finite difference method.

In this study, surface albedo is taken from the surface reflectance climatology at 440 nm which is derived from GOME-2 measurements from 2007 - 2013. The uncertainty of surface albedo ($A_s$) is assumed to be the difference between albedo derived at 425 nm and 440 nm to account for the small variation of albedo within the spectral fitting window. Information of surface





pressure ($P_s$) is taken from a digital elevation model (DEM) which is considered rather accurate and the uncertainty of surface pressure is mostly related to the variation within the GOME-2 footprint. We have analyzed this variation of surface pressure and find it is mostly (95 %) below 10 hPa. Therefore, we set the uncertainty of $P_s$ to 10 hPa.

The error related to a priori vertical distribution of water vapor is determined by using the a priori water vapor from the last iteration plus $1\sigma$ standard deviation which is also included in the look-up table. This new profile is then used to calculate the corresponding AMF. The difference between this AMF and the original AMF is taken as the uncertainty from the a priori profile. The uncertainty of the water vapor slant column can potentially affect the dynamic search of the a priori profile. As the slant column uncertainty can be much higher than the slant column itself over dry areas in the upper latitudes, considering this effect in the vertical profile uncertainty estimation would further amplify the uncertainty and results in unrealistic high

error. Therefore, we assume this effect is well covered by the vertical profile variation and accounted in the vertical profile uncertainty estimation. The error of the clear sky AMF can be calculated following Equation 11.

$$\sigma^2_{amf_{clr}} = \left( \frac{\partial AMF_{clr}}{\partial A_s} \sigma_{A_s} \right)^2 + \left( \frac{\partial AMF_{clr}}{\partial P_s} \sigma_{P_s} \right)^2 + \left( \frac{\partial AMF_{clr}}{\partial c_l} \sigma_{c_l} \right)^2 \tag{11}$$

where $\sigma_{amf_{clr}}$, $\sigma_{A_s}$, $\sigma_{P_s}$ and $\sigma_{c_l}$ are the uncertainty of the clear sky AMF, surface albedo, surface pressure, and water vapor profile, respectively. This error is in general <5 % for GOME-2 measurements over the tropics (30°S - 30°N).

### 3.1.10 Cloudy air mass factor error

The calculation of the uncertainty of the cloudy AMF is similar to the one used for the clear sky AMF, with surface albedo and surface pressure uncertainties replaced by cloud albedo and cloud top pressure errors. In this study, cloud top pressure

error is assumed to be 50 hPa (Theys et al., 2017; De Smedt et al., 2018). Previous studies show that the error of cloud albedo is compensated by the corresponding error of cloud fraction resulting a negligible net effect on trace gas retrieval (Van Roozendael et al., 2006; Lutz et al., 2016). Therefore, we assumed a cloud albedo uncertainty of 0.02 and intensity weighted cloud fraction uncertainty of 0.02. The combined effect of the assumed cloud albedo and cloud fraction uncertainties on water vapor retrieval is comparable to the assumption with just cloud fraction error of 0.05 (Theys et al., 2017; De Smedt et al., 2018). The error of

the cloudy AMF can be express as Equation 12.

$$\sigma^2_{amf_{cld}} = \left( \frac{\partial AMF_{cld}}{\partial A_c} \sigma_{A_c} \right)^2 + \left( \frac{\partial AMF_{cld}}{\partial P_c} \sigma_{P_c} \right)^2 + \left( \frac{\partial AMF_{cld}}{\partial c_l} \sigma_{c_l} \right)^2 \tag{12}$$

where $\sigma_{amf_{cld}}$, $\sigma_{A_c}$, $\sigma_{P_c}$ and $\sigma_{c_l}$ are the uncertainty of the cloudy AMF, cloud albedo, cloud top pressure, and water vapor profile, respectively. The error of the cloudy AMF ($\sigma_{amf_{cld}}$) in general varies from 25 % (25[th] percentile) to 40 % (75[th] percentile) for GOME-2 measurements over the tropics (30°S - 30°N).



### 3.1.11 Air mass factor error

Following Equation 8, the uncertainty of the total AMF can be derived from the clear sky and cloudy AMFs through error
propagation. The error of the total AMF can be calculated following Equation 13.

$$
\begin{aligned}
\sigma^2_{amf} = \quad & (AMF_{cld} \times CF_{iw})^2 \times \left( \left( \tfrac{\sigma_{amf_{cld}}}{AMF_{cld}} \right)^2 + \left( \tfrac{\sigma_{cf_{iw}}}{CF_{iw}} \right)^2 \right) \\
& + (AMF_{clr} \times (1 - CF_{iw}))^2 \times \left( \left( \tfrac{\sigma_{amf_{clr}}}{AMF_{clr}} \right)^2 + \left( \tfrac{\sigma_{cf_{iw}}}{1 - CF_{iw}} \right)^2 \right)
\end{aligned}
\tag{13}
$$

where $\sigma_{cf_{iw}}$ is the uncertainty of intensity weighted cloud fraction which is assumed to be 0.02 in the retrieval. The uncertainty of AMF ($\sigma_{amf}$) for GOME-2 measurements over the tropics (30°S - 30°N) varies in a range of 6 - 22 % ($25^{th}$ and $75^{th}$ percentile), while the error reduces to ∼6 % if the measurements are filtered for intensity weighted cloud fraction below 0.5. The uncertainty of AMF only shows a small latitudinal dependency on surface properties (albedo) and cloud patterns, observation and solar geometries. When all measurements are considered, the uncertainty of AMF varies from 8 % ($25^{th}$ percentile) to 24 % ($75^{th}$ percentile) with median value of 16 % while the mean error remains at ∼6 % for measurements with intensity weighted cloud fraction below 0.5.

### 3.1.12 Total error

Combining the slant column density error with the AMF error, the error of TCWV can then be calculated following Equation 9. The error of TCWV of GOME-2 measurements over the tropics (30°S - 30°N) is on average about 19 % under clear sky conditions (intensity weighted cloud fraction <0.5). A summary of the major sources of error in the water vapor retrieval in shown in Table 3.

### 3.1.13 Gridded total column water vapor

The ground pixels of the satellite observations vary in size and shape and often multiple pixels overlap in higher latitudes. To better reconstruct the spatial distribution of satellite observations and compare the results to different data sets, the retrieved GOME-2 water vapor columns are gridded onto a high resolution latitude longitude grid with a spatial resolution of 0.02° × 0.02°. The gridded data is based on all valid vertical columns within a certain period, i.e., a day or a month. Valid measurements are defined with corresponding solar zenith angle smaller than 85°, intensity weighted cloud fraction smaller than 0.5, root mean square of spectral fit residual less than 0.002, and AMF larger than 0.1. The vertical column of each valid





**Table 3.** Summary of the major sources of error in the water vapor retrieval.

| Error | Type | Typical uncertainty |
|---|---|---|
| Instrument noise | SCD | 14 % |
| Absorption cross section | SCD | 3 % |
| Surface albedo | AMF | 2 % |
| Surface pressure | AMF | 1 % |
| Cloud fraction | AMF | 3 % |
| Cloud albedo | AMF | 3 % |
| Cloud top pressure | AMF | 3 % |
| A priori profile | AMF | 10 % |

pixel is stored in all grid points lying within the satellite ground pixel boundaries. These pixel boundaries are taken from the level 1B data. For overlapping pixels, a weighted average is calculated where the weighting is defined by Equation 14.

$$VCD_g = \frac{\sum VCD_i \times w_i}{\sum w_i} \text{ with } w_i = \frac{1}{A \times (1 + 3 \times CF_{iwi})^2} \tag{14}$$

where VCD$_g$ is the gridded water vapor column while VCD$_i$ represents each individual measurement. The weighting is denoted as $w$ which is dependent on the intensity weighted cloud fraction ($CF_{iw}$) and GOME-2 ground pixel size ($A$). As clear sky data are more reliable, the gridding scheme gives higher weights to clear sky pixels. It is recommended to give higher weights to smaller pixels to enhance the fine details in the gridded product (Wenig et al., 2008; Chan et al., 2012). Since the ground pixel size of the GOME-2 backward scan is 3 times larger than the forward scan, the gridded data are mainly weighted toward the forward scan. Examples of daily, monthly, and seasonal average of GOME-2A observations of TCWV are shown in Figure 6.

## 3.2 Comparison methods

In this study, water vapor columns retrieved from GOME-2 observations in the blue band are compared to ground based sun-photometer and radiosonde measurements. As the satellite, sun-photometer and radiosonde data are different in spatial and temporal resolution and coverage, only coinciding data are used in the comparison. The criteria to select coinciding data are (1) satellite data are selected such that the center coordinate of the satellite pixel is within 50 km of the sun-photometer or radiosonde site, (2) sun-photometer or radiosonde data are selected around the satellite overpass time such that the time difference between the satellite and ground observations is less than 2 hours. Subsequently satellite, sun-photometer and radiosonde measurements are averaged to daily data for comparison. As sun-photometer only provides data under clear sky conditions, satellite data are filtered for intensity weighted cloud fraction smaller than 0.5 for consistency. Daily averaged GOME-2 data are used for the comparison to sun-photometer and radiosonde measurements.



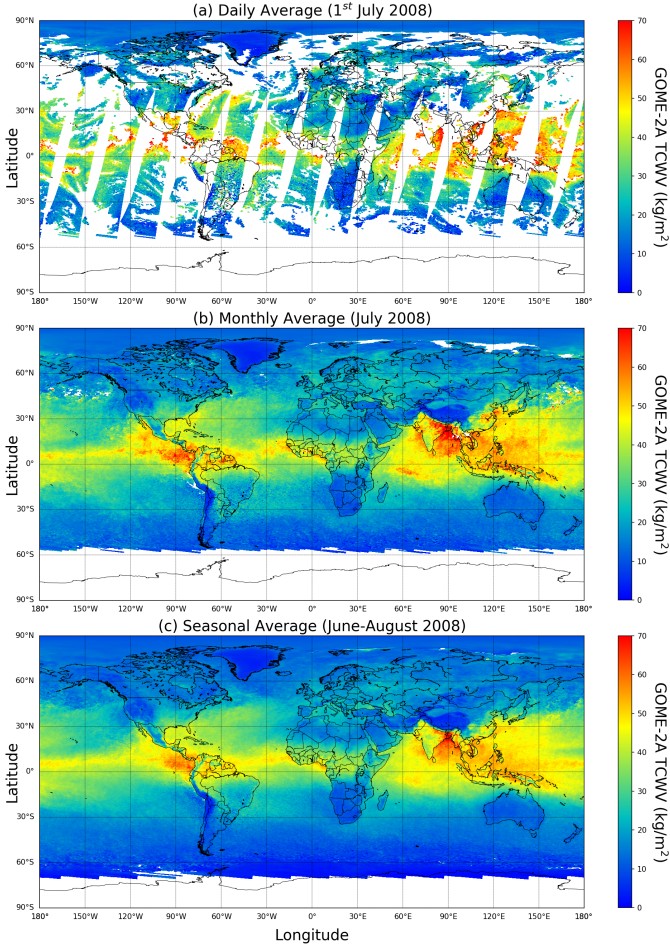

**Figure 6.** (a) Daily ($1^{st}$ July 2008), (b) monthly (July 2008), and (c) seasonal (June - August 2008) average of GOME-2A observations of TCWV.

## 4 Results

In this section, we present validation studies of GOME-2 TCWV retrieved in the blue spectral range. Our retrieval results are compared to the GOME-2 operational water vapor product which is derived in the red spectral band. In addition, the new data set is validated against ground based sun-photometer observations and radiosonde measurements.



**Figure 7.** Monthly average TCWV derived from the GOME-2A observations. Top panels show data from the blue band retrieval, center panels show data from the GOME-2 operational product (red band) and bottom panels show the differences between the two data sets. The left panels show the data of January 2018, while the right panels show the data of July 2018.

## 4.1 Comparison to the GOME-2 operational product

### 4.1.1 Spatial distribution comparison

Figure 7a and d shows the monthly average spatial distribution of TCWV retrieved from GOME-2A observations in the blue spectral band for January and July 2018. These months are chosen as examples for winter and summer. TCWV from the

5    GOME-2 operational product are shown in Figure 7b and e for comparison. Both data sets are gridded and filtered in the same way as described in Section 3.1.13. Missing data over the Tibet Plateau and Andes Mountains are due to no valid data available





over high altitude areas in the operational product, while missing data over other smaller regions are mainly related to cloud filtering. The differences between the two data sets are plotted in Figure 7c and f. Both data sets show very similar spatial patterns with higher water vapor columns over the tropics and lower values at upper latitudes. The blue band retrieval shows significantly higher water vapor columns over west Africa ($\sim$5 kg/m$^2$), India ($\sim$7 kg/m$^2$) and Indochina Peninsula ($\sim$6 kg/m$^2$)

during summertime of the northern hemisphere. In addition, the blue retrieval shows a small negative bias of about 0.5 kg/m$^2$ over oceans in the tropics.

### 4.1.2   Zonal average, correlation and bias

The water vapor columns derived from the blue band and the operational retrieval are sorted by their measurement latitudes and plotted in Figure 8a and c. Data from January and July of 2018 are shown. The retrieval in the blue spectral band show good

zonal agreement with the operational product in both winter and summer. The 1 $\sigma$ standard deviation variation ranges of both data sets overlap with each other, indicating both data sets capture similar spatial variations of water vapor columns. Direct comparison of individual measurements from both data sets is shown in Figure 8b and d. The two data sets show very good agreement with Pearson correlation coefficients ($R$) ranging from 0.91 up to 0.94. The correlation is slightly better during winter (January) of the northern hemisphere. The mean bias between the blue band retrieval and the operational product is

0.12 kg/m$^2$ in January 2018 and -0.08 kg/m$^2$ in July 2018. Although the differences between the two data sets are small, it is still statistically significant ($P$ value $\ll$0.01) due to the large number of sample used in the comparison.

Figure 9 shows the monthly zonal averaged TCWV derived from GOME-2A measurements for 11 years from 2008 to 2018. Both the blue retrieval and operational data sets are shown. Water vapor columns from both data sets show very similar zonal distribution patterns. Compared to the operational product, the blue retrieval before 2015 shows slightly lower water vapor

columns over the tropics and higher values in the upper latitudes and resulting a small wet bias of $\sim$1 kg/m$^2$. The wet bias is greatly reduced to less than 0.1 kg/m$^2$ after 2015.

Figure 10 shows the time series of correlation and mean bias between the blue retrieval and the operational algorithm. Data from 2008 to 2018 are shown. The correlation between the two data sets are in general very good, with Pearson correlation coefficient ($R$) ranging from 0.90 to 0.96. The correlation between both data sets is generally higher in winter of the northern

hemisphere and lower during summer. Significant overestimation of TCWV is observed for measurements from 2008 to 2015. The bias between the two data sets is greatly improved after 2015.

In addition, we have compared the TCWV measured by both GOME-2A and GOME-2B to investigate the cross sensors consistency. The mean water vapor column retrieved from GOME-2A observations from 2013 to 2014 is 20.72 kg/m$^2$, while GOME-2B observations show a similar value of 20.91 kg/m$^2$. The bias between the two GOME-2 sensors before the level 1B

data update is $\sim$1 %. The mean water vapor column retrieved from GOME-2A observations from 2016 to 2018 is 20.53 kg/m$^2$, while GOME-2B shows a very similar value of 20.87 kg/m$^2$. The bias of water vapor column retrieved in the blue band between the two sensors remains at a similar level (<2 %) after the update of level 1B data. On the other hand, the bias of water vapor column retrieved in the red band between GOME-2A and GOME-2B lies between 3 - 4 % for data before and after the level 1B update.

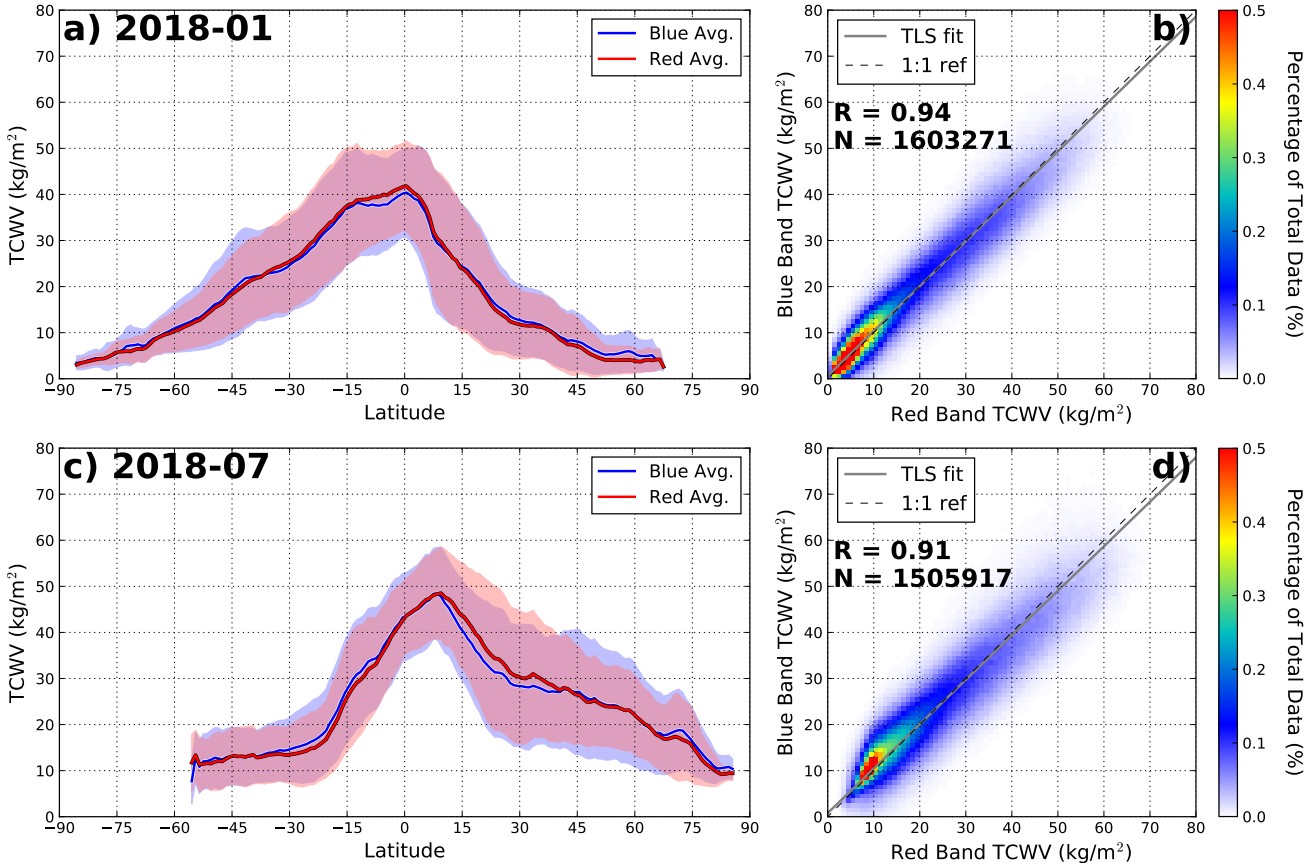

**Figure 8.** Comparison of TCWV from the blue retrieval and operational algorithm. (a) Zonal and (b) direct comparison in January 2018. Lower panels are the same as the upper panels but for July 2018. Shadowed areas in the zonal average plots indicate the $1\sigma$ standard deviation variation range. The color code of the scatter plots indicates the relative portion of total measurement pixels. Individual GOME-2A measurements are used in the comparison.

### 4.1.3 Long term variations

Figure 11a and c shows the time series of annual mean TCWV derived from GOME-2A and GOME-2B observations. The rate of change of TCWV calculated from GOME-2A and GOME-2B measurements is also shown in Figure 11b and d, respectively. Both GOME-2A blue and red band measurements in general suggest a slightly increasing trend. The inter-annual variation of TCWV captured by the blue band retrieval and operational product agrees well with each other except the year of 2015 when the level 1B processor was updated. The averaged rate of change of TCWV derived from GOME-2A by the blue retrieval is about $0.12\,\text{kg/m}^2$ per year, while a higher increasing rate of $0.19\,\text{kg/m}^2$ per year is observed in the red band. If we remove the year of 2015 from the analysis, the average rate of change calculated from the blue retrieval increases from $0.12\,\text{kg/m}^2$ per



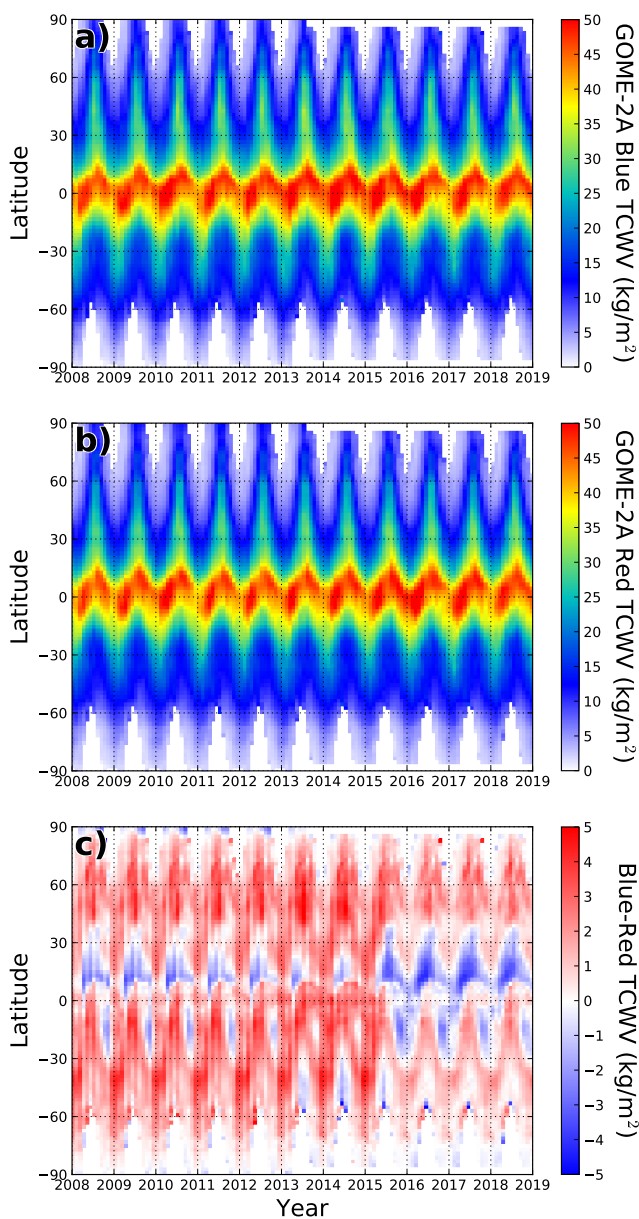

**Figure 9.** Monthly zonal average of TCWV from GOME-2A observations in (a) blue and (b) red spectral bands. The differences between the two data sets are shown in (c). Data over 11 years from 2008 to 2018 is shown.

year to $0.17\,\mathrm{kg/m^2}$ per year, and agrees better with the operational product. A similar increasing rate can also be observed by the GOME-2B sensor, with an averaged rate of change derived from the blue and red band of 0.12 and $0.21\,\mathrm{kg/m^2}$ per year,





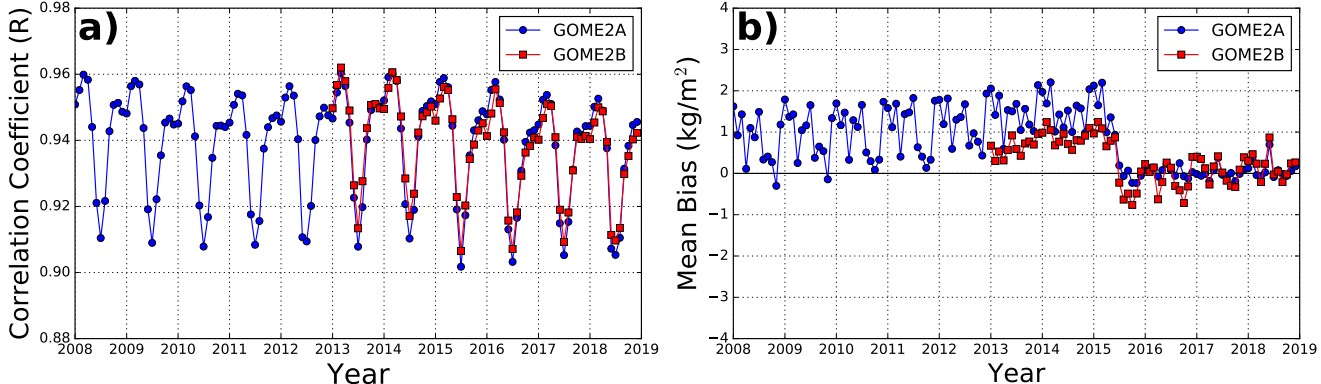

**Figure 10.** Time series of Pearson correlation coefficient between water vapor columns from the blue band retrieval and operational algorithm is shown in (a). (b) shows the mean bias between the two data sets. Both GOME-2A and GOME-2B data are shown. Individual measurements are used in the calculation of correlation coefficient and mean bias.

respectively. If the year of 2015 is removed from the trend analysis, then the increasing rate derived from the blue band would increase to $0.26\,\text{kg/m}^2$ per year. Although the increase rate of $0.12$ - $0.26\,\text{kg/m}^2$ per year is not significant compared to the typical temporal variation of water vapor ($\sim 2.5\,\text{kg/m}^2$), the trend of atmospheric water vapor content is a major concern of climate change and has to be cross validate with other observations and model simulations to investigate the causes and the impacts to the climate system. A further discussion of this topic is however beyond the scope of this study.

### 4.2 Comparison to sun-photometer data

Figure 12a and b shows the scatter plot of GOME-2A and GOME-2B measurements of TCWV against sun-photometer measurements. The selection criteria for data sets used in the comparison are presented in Section 3.2. Co-located daily average data are used in the comparison. The sun-photometer and GOME-2 measurements of TCWV agree well with each other. The Pearson correlation coefficient ($R$) is 0.91 and 0.89 for GOME-2A and GOME-2B observations, respectively. The slope of the total least squares regression line for GOME-2A comparison is 0.99 with an offset of $0.84\,\text{kg/m}^2$. The analysis of GOME-2B data shows a similar result with slope of 1.00 and offset of $1.03\,\text{kg/m}^2$. The mean bias between sun-photometer data and observations from GOME-2A and GOME-2B is $0.78\,\text{kg/m}^2$ and $1.09\,\text{kg/m}^2$, respectively.

Figure 13 shows the statistic of the differences between sun-photometer and GOME-2 measurements of TCWV. Data are sorted by year, month, and latitude, to investigate the spatio-temporal agreement between the two data sets. The interannual variation analysis shows a small positive bias of $1$ - $2\,\text{kg/m}^2$ for both GOME-2A and GOME-2B observations before 2015. The overestimation is significantly improved after the update of level 1B data in 2015. The discrepancies between GOME-2 and sun-photometer show a larger variation range in summer months of the northern hemisphere. In addition, larger variation of discrepancies is also observed over the tropics compared to upper latitudes.





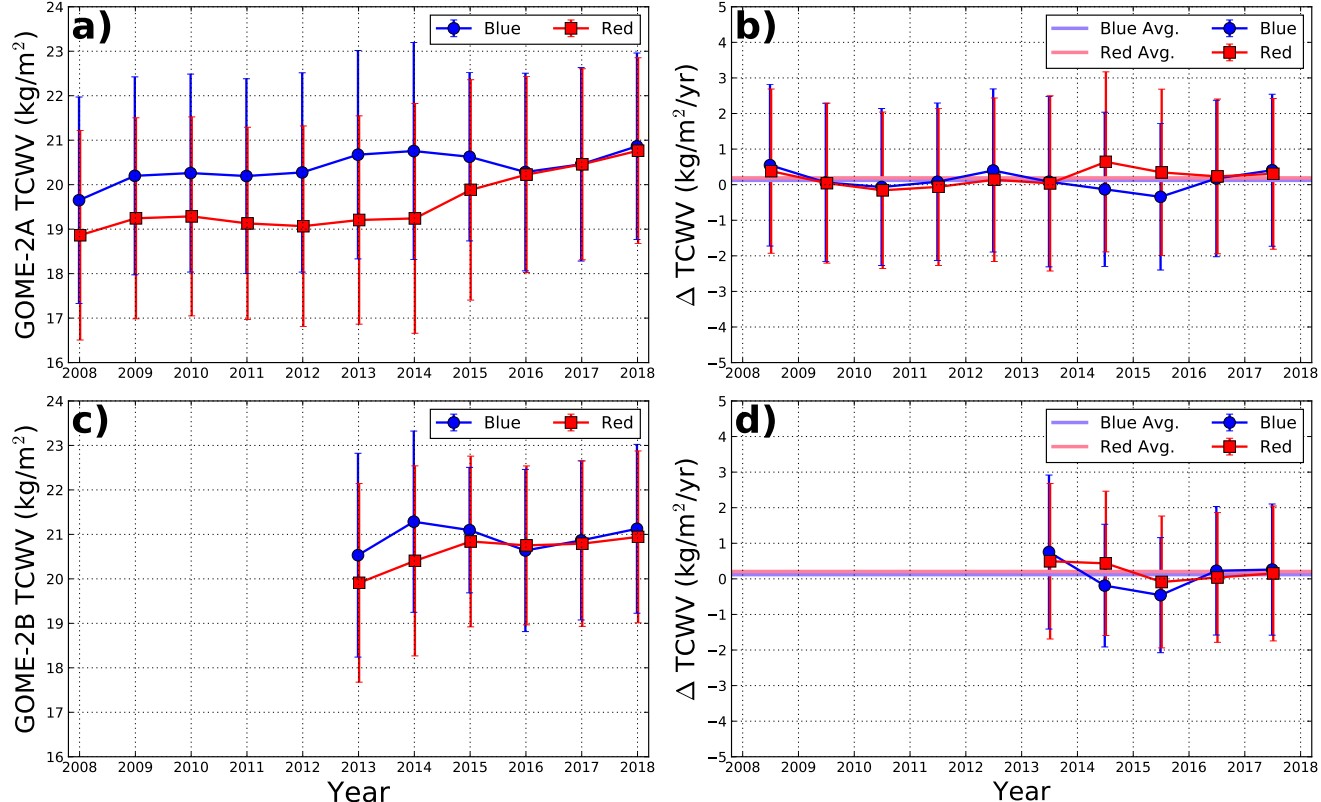

**Figure 11.** Time series of annual average of TCWV retrieved from (a) GOME-2A and (c) GOME-2B in blue (blue curves) and red (red curve) spectral bands. The rate of change of TCWV derived from (b) GOME-2A and (d) GOME-2B are also shown. The purple and pink lines indicate the average rate of change derived from the blue and red band measurements, respectively. The error bars indicate the $1\,\sigma$ standard deviation of the annual variation.

### 4.3 Comparison to radiosonde measurements

The scatter plots of the radiosonde TCWV measurements to GOME-2A and GOME-2B measurements are shown in Figure 14. The selection criteria for data sets used in the comparison are presented in Section 3.2. Co-located daily average data are used in the comparison. Both GOME-2A and GOME-2B measurements are consistent with the radiosonde measurements with Pearson correlation coefficient ($R$) of 0.92 and 0.91, respectively. The slope of the total least squares regression line for GOME-2A comparison is 0.99 with an offset of $1.33\,\mathrm{kg/m^2}$. A similar agreement can also be obtained from GOME-2B observations, with a slope of 1.02 and offset of $0.42\,\mathrm{kg/m^2}$. The mean bias between radiosonde data and observations from GOME-2A and GOME-2B are $1.20\,\mathrm{kg/m^2}$ and $0.88\,\mathrm{kg/m^2}$, respectively.





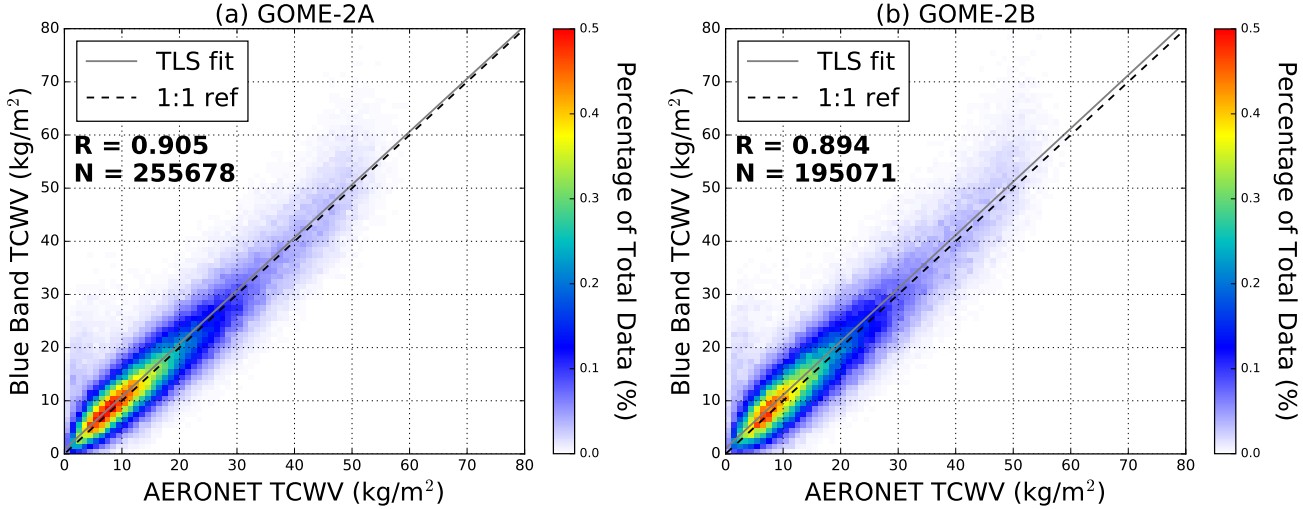

**Figure 12.** Comparison of TCWV measured by sun-photometer to GOME-2A is shown in (a) while the comparison to GOME-2B is shown in (b). Co-located daily average data are used in the comparison.

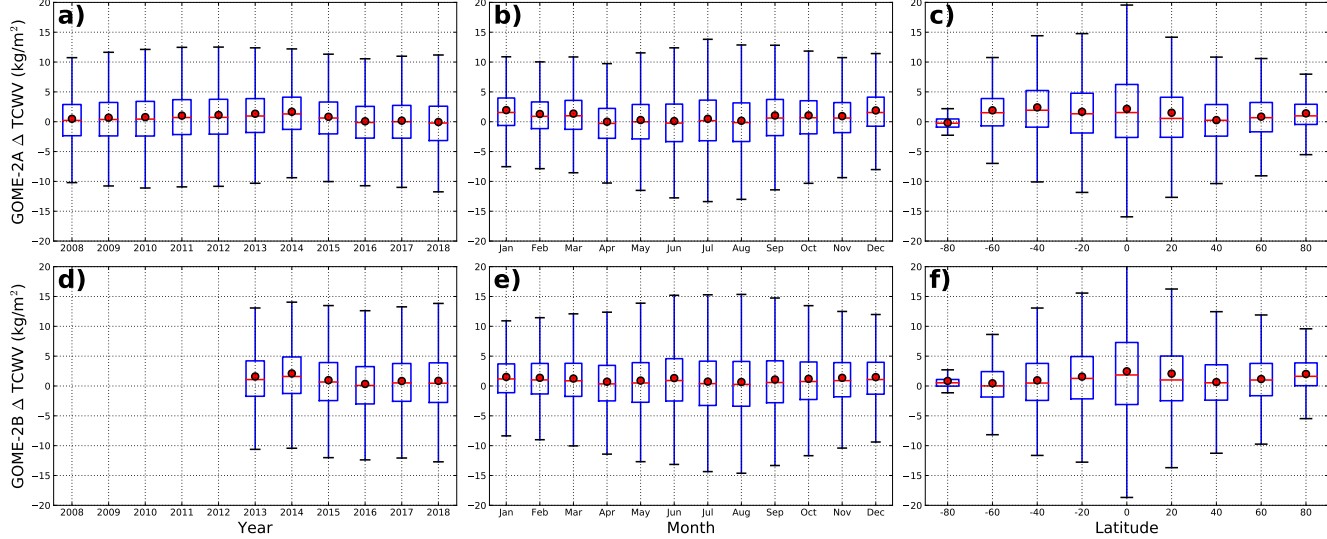

**Figure 13.** Comparison between sun-photometer and GOME-2A observations. Top panels show GOME-2A data and bottom panels show GOME-2B data. Data are sorted by year (the left column), month (the middle column) and latitude (the right column).

Figure 15 shows the statistic of the differences between radiosonde and GOME-2 measurements of TCWV. Data are sorted by year, month, and latitude, to investigate the spatio-temporal agreement between the two data sets. Similar to the sun-





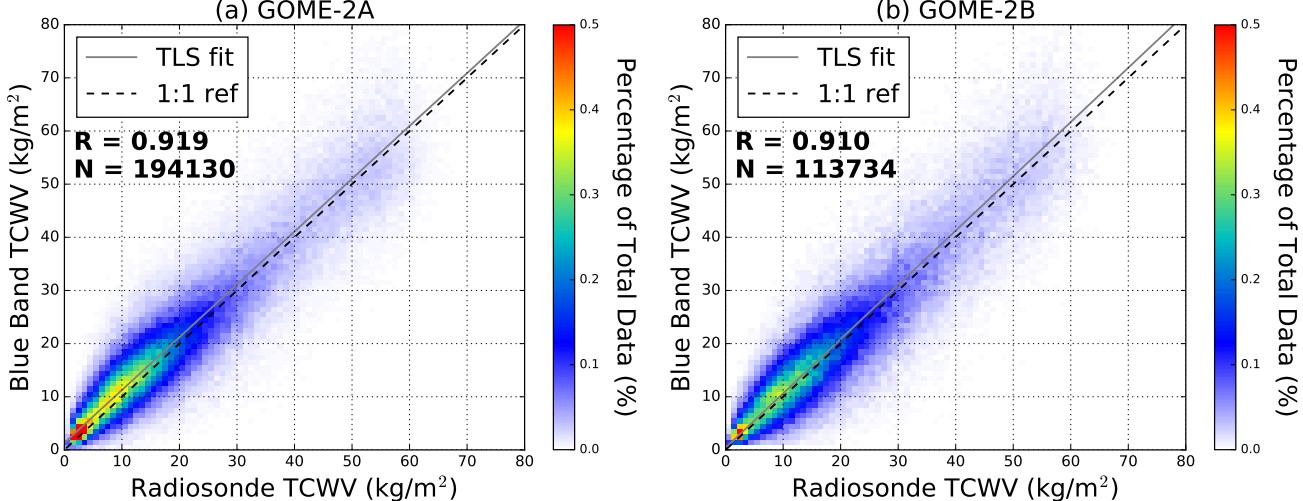

**Figure 14.** Comparison of radiosonde measurements of TCWV to GOME-2A is shown in (a) while the comparison to GOME-2B observations is shown in (b). Co-located data are used in the comparison.

photometer comparison result, the GOME-2 data overestimated the water vapor columns by ∼1 kg/m$^2$ before 2015. The monthly pattern also shows a larger variation range in summer months of the northern hemisphere. The discrepancies between GOME-2 and radiosonde also vary in a larger range over the tropics and lower at upper latitudes.

# 5 Discussion

## 5.1 Comparison to the GOME-2 operational product

### 5.1.1 Spatial distribution comparison

The spatial distribution of water vapor from the blue and operational retrieval shows good consistency with each other. However, the blue retrieval show significantly higher values over west Africa, India and Indochina Peninsula and slightly lower values over oceans in the tropics in July. Previous study reported that the operational GOME-2 product is underestimating water vapor columns over land and overestimating over oceans in the tropics (Grossi et al., 2015). The differences between the two data sets indicate that the blue band retrieval improved the bias over these areas.

Overestimation of TCWV can also be observed over south America in both summer and winter. The discrepancies are likely related to the uncertainties of Lambertian assumption of surface albedo over vegetation. The bidirectional reflectance distribution function (BRDF) effect has been reported to have significant impacts on the retrieval of cloud and trace gas over forested scenes (Lorente et al., 2018). The uncertainty of cloud product due to the BRDF effect over vegetation also indirectly

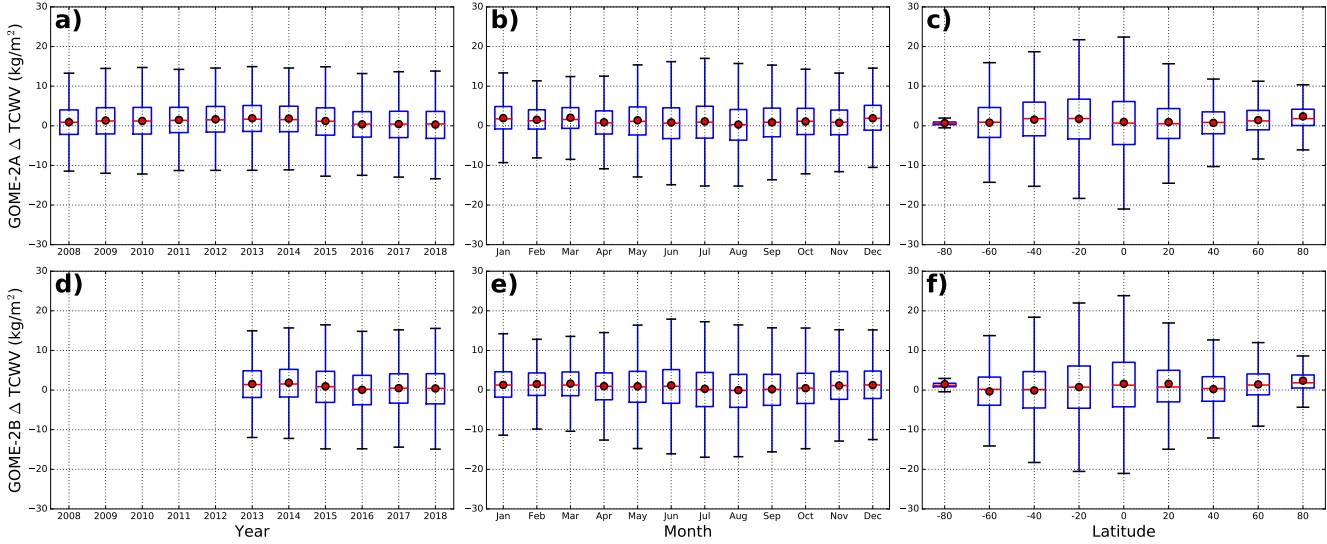

**Figure 15.** Comparison between radiosonde and GOME-2A observations. Top panels show GOME-2A data and bottom panels show GOME-2B data. Data are sorted by year (the left column), month (the middle column) and latitude (the right column).

affects the water vapor retrieval. In this study, the Lambertian surface assumption is used in the water vapor retrieval to be consistent with the cloud product. Using the Lambertian surface assumption over areas covered by vegetation probably leads to overestimation of the water vapor columns. In addition, inter-annual variation of surface also affects the retrieval results. Sütterlin et al. (2016) analyzed the Advanced Very High Resolution Radiometer (AVHRR) BRDF product from 1990 - 2014

5 and the result shows the inter-annual variability of the land surface albedo is in general less than 0.01 for snow-free vegetation cover but possibly larger than 0.06 for regions covered by snow or ice. We performed a sensitivity analysis to quantify the uncertainty TCWV caused by inter-annual variation of albedo by using the numbers provided in Sütterlin et al. (2016). The result shows that the uncertainty of TCWV due to inter-annual variations of surface albedo is in general $<2\%$ while the uncertainty increased to $\sim9\%$ for areas covered by snow and ice. In the future, we plan to update the surface albedo retrieval

10 to account for the temporal variation of abldeo and the BRDF effect (Loyola et al., 2019). The updated surface albedo product will also improve the accuracy of the cloud retrieval and further improve the water vapor retrieval.

### 5.1.2 Zonal average, correlation and bias

Compared to the operational product, the blue retrieval is overestimating the water vapor columns at upper latitudes and underestimating in the tropics and resulting a small overestimation ($\sim5\%$) for measurements before 2015. Water vapor columns

15 retrieved at the blue band are in general reduced in both tropical regions and high latitudes after 2015. As a result, the difference between the two data sets becomes smaller at higher latitudes while a slightly stronger underestimation is observed over the tropics. This change is likely related to the level 1B data processing switched from version 6.0 to version 6.1 on $25^{th}$ June





2015, which affects the blue band but not the red band. Although the uncertainty caused by the contaminated level 1B data is small and well covered by the assumed uncertainties, the bias is still significant when averaging large number of data for climate studies. The overall bias between the two GOME-2A data sets is reduced from $1.14 \, \text{kg/m}^2$ before the update (2008 - 2014) to $0.05 \, \text{kg/m}^2$ after the update (2016 - 2018). A similar effect is also observed in the GOME-2B water vapor retrieval.

The mean bias between the GOME-2B blue retrieval and operational product is $\sim 0.75 \, \text{kg/m}^2$ before the update (2013-2014) and is reduced to $\sim 0.05 \, \text{kg/m}^2$ after the update (2016 - 2018). The result indicates that reprocessing of level 1B data before 2015 is necessary to produce reliable TCWV data set.

Previous comparison of the GOME-2 operational water vapor product to radiosonde measurements shows that the operational product is on average underestimating water vapor columns over land by $1.0 \, \text{kg/m}^2$ and overestimating over ocean by

$1.5 \, \text{kg/m}^2$ (Kalakoski et al., 2016). After the update of level 1B data in 2015, the blue retrieval is reporting slightly higher water vapor columns than the operational product at upper latitudes, and lower values are observed over the tropics.

## 5.2 Comparison to sun-photometer data

The small positive offsets between GOME-2 and sun-photometer measurements indicates that the blue band retrieval slightly overestimates the TCWV. On the other hand, the sun-photometer data has been reported to underestimate TCWV by 6 - 9 %

compared to GPS data (Pérez-Ramírez et al., 2014). In addition, the GOME-2 to sun-photometer comparison also includes data before 2015, where the level 1B data are contaminated and enhance the TCWV by up to $2 \, \text{kg/m}^2$. If we only consider data taken after 2015 in the comparison, the bias of GOME-2A would reduced from $0.78 \, \text{kg/m}^2$ to $0.09 \, \text{kg/m}^2$ and the bias of GOME-2B also reduced from $1.09 \, \text{kg/m}^2$ to $0.64 \, \text{kg/m}^2$. Considering all the uncertainty of the sun-photometer (6 - 9 %) and GOME-2 ($\sim 20 \, \%$) measurements, the small discrepancies between the two data sets are considered reasonable.

The analysis of bias between GOME-2 and sun-photometer measurements shows a larger variation during summer months of the northern hemisphere. This is partly related to the geolocation distribution of the sun-photometer stations. Most of the stations are situated in the northern hemisphere and result in larger number of valid measurements and variation in summer. In addition, both GOME-2A and GOME-2B are slightly overestimating the water vapor columns by $\sim 1 \, \text{kg/m}^2$ during winter months, i.e., January, February and December. Our observations are consistent with the previous radiosonde comparison study

(Antón et al., 2015). The reason of larger discrepancies during winter is likely related to the variation of surface albedo (snow and ice cover). The zonal variation analysis shows larger variations in the tropics while the variations are much smaller at higher latitudes. The uncertainty of TCWV derived from satellite observations is strongly related to the air mass factor uncertainty. As the air mass factor is a multiplication term and there is larger amount of water vapor over the tropics, resulting larger absolute uncertainty.

## 5.3 Comparison to radiosonde measurements

The small overestimation of water vapor columns by GOME-2 compared to radiosonde measurements is partly related to the level 1B data issue before 2015. If we only consider data taken after 2015 in the comparison, the overestimation of GOME-2A is reduced from $1.20 \, \text{kg/m}^2$ to $0.36 \, \text{kg/m}^2$ and the bias of GOME-2B is also reduced from $0.88 \, \text{kg/m}^2$ to $0.31 \, \text{kg/m}^2$. In





addition, the radiosonde measurements stop at a certain altitude and do not cover the entire atmosphere, which may slightly underestimate the total column. The discrepancy between the satellite and radiosonde measurements is below 5 %, which is well within the uncertainties of radiosonde measurements reported from previous studies (Wang and Zhang, 2008; Van Malderen et al., 2014). Previous comparison of the GOME-2 operational product to radiosonde data shows a dry bias of 4 - 11 % Antón et al. (2015). In contrast, the new retrieval results show a much more reasonable wet bias of <2 %. Considering the uncertainty radiosonde and the GOME-2 retrieval, the two data sets are in good agreement with each other.

## 6  Conclusions

In this work, we have developed a water vapor retrieval algorithm in the visible blue band of 427.7 - 455 nm, providing an alternative solution for satellite sensors that do not cover the red band where TCWV is typically retrieved. The major advantage of the new water vapor retrieval algorithm is that it does not rely on a priori information from a chemistry transport model. This improvement makes the satellite product independent from model simulations and avoids model errors propagating to the measurement, making the data more suitable for climate studies.

The developed TCWV retrieval has been successfully applied to GOME-2. Water vapor columns retrieved in the blue band show very good spatio-temporal consistency with the operation product, sun-photometer and radiosonde measurements. However, reprocessing of GOME-2 level 1B data before 2015 is necessary to produce a reliable climate record. The blue band retrieval results are consistent between GOME-2A and GOME-2B with discrepancies of less than 2 %. The retrieval is feasible to be applied to former, current, and forthcoming UV and Vis satellite sensors to create an independent water vapor climate data record starting from 1995 and continuing for the next two decades.

*Competing interests.* The authors declare that they have no conflict of interest.

*Acknowledgements.* The work described in this paper was supported by the German Aerospace Center (DLR) programmatic.



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
