# Peer review of "Total column water vapor retrieval for GOME-2 visible blue observations"

_Atmospheric Measurement Techniques, 2020_

## Referee Comment (RC1) · Anonymous Referee #1 · 26 Mar 2020

A new total column water vapor (TCWV) retrieval algorithm in the visible blue spectral band for the Global Ozone Monitoring Experience 2 (GOME-2) instruments on board the EUMETSAT MetOp satellites is presented. The blue spectral band algorithm is important because it allows retrieval of water vapor from sensors which do not cover longer wavelengths, such as Ozone Monitoring Instrument (OMI) and the Copernicus atmospheric composition missions Sentinel-5 Precursor (S5P), Sentinel-4 (S4) and Sentinel-5 (S5). This algorithm applies the differential optical absorption spectroscopic (DOAS) technique to retrieve water vapor slant columns and has an iterative optimization module to dynamically find the optimal a priori water vapor profile. This method is better suited for climate studies than usual satellite retrievals with static a priori or vertical profile information from chemistry transport model (CTM). The dynamic a priori

algorithm makes use of the fact that the vertical distribution of water vapor is strongly correlated to the total column. The new algorithm is applied to GOME-2A and GOME-2B observations to retrieve TCWV. The validation of this data set includes comparisons to the operational product retrieved in the red spectral band from sun-photometer and radiosonde measurements. These TCWV retrieved in the blue band, which are in good agreement with the other data sets, indicate that the new algorithm derives precise results and can be used for the current and forthcoming Copernicus Sentinel missions S4 and S5. General comments The main motivation for this work is to develop a new water vapor retrieval in the available spectral bands of satellite observations. Further, and this is formulated as requirement also, the retrieval should not rely on input from chemistry transport model (CTM) to avoid propagating model error into the climatological measurement records. Following this, a detailed error estimation is performed. To do so, also water vapor columns in the standard AERONET product as well as temperature, relative humidity, dew point depression, wind direction and wind speed data at multiple pressure levels from radiosondes from stations all over the world are used together with ERA-Interim reanalysis data of water vapor vertical distribution and the relation to their total column amount. The paper addresses relevant scientific questions within the scope of AMT. It completes the knowledge about long-term water vapor content variation of the atmosphere. The paper presents novel concepts, ideas and tools. The scientific methods and assumptions are valid and clearly outlined so that substantial conclusions are reached. The description of experiments and calculations are sufficiently complete and precise to allow their reproduction by fellow scientists. The quality and information of the figures is fine. The related work is well cited as well as the number and quality of references appropriate i.e. the authors give proper credit to related work and clearly indicate their own new/original contribution. The title and the abstract clearly reflects the contents of the paper. The overall presentation is well structured and clear. The language is fluent and precise. The mathematical formulae, symbols, abbreviations, and units are generally correctly defined. Specific Comments It would be helpful to describe in the beginning of section 2 "Instruments and data sets"

to describe the overall evaluation strategy – why these data described here are applied? Technical corrections Not all references include the doi number. It should be unique.

---

## Referee Comment (RC2) · Anonymous Referee #2 · 21 May 2020

The paper titled 'Total column water vapor retrieval for GOME-2 visible blue observations' by Chan et al. presents a new method to retrieve total column water vapor (TCWV) from GOME-2 spectra in the blue wavelength band. Slant column water vapor is retrieved using the DOAS spectral fitting technique, and subsequently converted to vertical column using Air Mass Factor (AMF). A dynamic search approach is used to find suitable a priori profile for the AMF calculation. The new algorithm is then applied to retrieve TCWV from GOME-2A and GOME-2B observations. TCWV results from GOME-2A and GOME-2B are also validated against GOME-2 measurements in the red band, sun-photometer and radiosonde measurements. The comparisons show that new blue band retrieval in general shows good agreement with other data sets, and proofed the reliability of the new algorithm. In general, this manuscript presents

an interesting results. However, there are still some concerns need to be addressed before publication in AMT.

1)The author introduced the source of a priori water vapor vertical profile in section 3.1.4, which is the statistical result of ERA reanalyzing data for 10 years. In section 2.6, it explained the horizontal and time resolution of ERA data but did not explain the vertical resolution. The vertical resolution and the number of layers should be clearly explained here, because the layer of a priori profile is used to calculate the AMF.

2)The third line on page 13 of the author indicates that "using a priori profile from model is not optimal for our water vapor retrieval", but the ECMWF ERA data used by the author is also the result of numerical simulation. Here the author only needs to emphasize that the a priori profile used is statistical data, which can reduce the model error.

3)The author wrote in the third line on page 16 that the profile below cloud is taken from the a priori profile. The water vapor below cloud is close to the atmospheric boundary layer, where the water vapor concentration is high and the change is large, so this approximation will produce a large error. In addition, should the thickness of the cloud layer also be considered?

4)The authors mentioned the level 1B issue appears at wavelength larger than 460 nm, and that is why they avoided including longer wavelength in the spectral analysis. However, we still see a jump of TCWV in 2015 when switching the level 1B data version. The authors should explain why this is still happening even they did not use wavelength longer than 460 nm in their analysis.

5) The details of the spectral fitting settings are scattered in the text which is quite difficult for the audience to follow. I would suggest the authors to summarize this information in a table.

6) The comparison of TCWV to the retrieval in the red band shows a positive bias over

vegetations, e.g., South America and Central Africa, which the authors claim that this is related to the uncertainty of the surface albedo data set. The authors also mentioned that they are trying to improve the albedo database to by using new surface albedo retrieval. It would be nice to show some preliminary results (if available) to show the new method can potentially reduce this bias.

7) Page 23, line 15-16: When the authors compare big data sets (sample size > 10k), even a small bias is significant, so I think there is no need and uncommon to mention the P value.

---

## Referee Comment (RC3) · Anonymous Referee #3 · 28 May 2020

The manuscript presents a new total column water vapor retrieval algorithm in the visible blue spectral band for the GOME-2 satellite instruments. The blue band algorithm has an advantage over the traditional red band algorithm that allows retrieval of water vapor from sensors which do not cover longer wavelengths, such as OMI and TROPOMI. One of the new features of the algorithm is the dynamic optimization of the a priori water vapor profile, which make use of the fast that the vertical distribution of water vapor is strongly correlated with the total columns. So the retrieval does not dependent on a priori information from model forecast and better suit for climate study. This paper presented a novel method to retrieve TCWV from satellite observations. Overall, the objective of the study is clear, and the results are validated properly. The presentation of the paper is well structured and easy to follow. Therefore, I suggest

publishing the paper after clarifying the minor issues listed below.

1) Please add a summary of the evaluation strategy in section 2 describing why these data are used. 2) The Level 1 data issue is still not clear to me, usually satellite measurement quality is decaying over time, why the data after 2015 is fine but the data before 2015 is contaminated? Is there any recalibration done in 2015? If the recalibration improves the data quality, will there be any reprocessing? 3) The error due to surface albedo listed in table 3 is 3%, while the bias over vegetation, e.g., Amazon and Central Africa, seems to be much larger. Please clarify. 4) Why the absorption cross section of NO2 is not ∼290k 5) Some of the references are inconsistence in format.

---

## Short Comment (SC1) · 10 Jun 2020

Chan et al. present a very interesting TCWV retrieval for GOME-2 which not only makes use of the $H_2O$ absorption in the blue spectral range, but also applies an iterative a priori water vapour profile approach. The authors are probably aware that Borger et al. (2020) developed a very similar approach. I have a few comments/questions that can hopefully help to further improve the overall high quality of the paper:

1. One great advantage of using the blue spectral range compared to the red spectral range is the higher sensitivity for the near-surface layers over ocean. Did the authors also consider to compare their TCWV data set to microwave satellite sensors? For instance SSMI/SSMIS are widely considered as reference measurements for TCWV

retrievals over ocean because they can measure under all-sky conditions. It would be very important to see how good the TCWV from GOME-2 can match the TCWV from SSMI/SSMIS.

2. A similar question is related to the use of AERONET data as reference. As the authors mention it themselves, TCWV from AERONET is potentially affected by biases. Have the authors had a look at TCWV from ground-based GPS measurements from SuomiNet or IGS? In comparison to AERONET, these GPS networks can conduct TCWV retrievals for all-sky conditions at a very high accuracy and provide continuous time series of TCWV for their measurement stations (e.g. SuomiNet TCWV data are available every 30 min). Such a comparison would further improve the confidence of the new retrieval.

3. It is kind of surprising to see that the GOME-2 TCWV retrieval uses HITEMP2010 / HITRAN2012 as H2O cross-section, whereas Wang et al. (2019) (for OMI) and Borger et al. (2020) (for TROPOMI) found a significantly better agreement to reference measurements by using HITRAN2008. The use of HITRAN2008 over HITRAN2012 is also supported by the LP-DOAS results in Lampel et al. (2015) (see Table 8 in their paper). What is the rationale for still using HITRAN2012?

4. Although it is very reasonable to use ERA-Interim for the statistical analysis, isn't there potentially the risk that the data quality of the ERA-Interim water vapour profiles can vary a lot e.g. depending on which measurement data have been used during the data assimilation process? At least this was one of the reasons for Borger et al. (2020) to only use water vapour profiles from a consistent measurement data set (COSMIC in this case) for setting up their iterative a priori water vapour profile retrieval scheme.

References

Borger, C., Beirle, S., Dörner, S., Sihler, H., and Wagner, T.: Total column water vapour retrieval from S-5P/TROPOMI in the visible blue spectral range, Atmos. Meas. Tech., 13, 2751–2783, https://doi.org/10.5194/amt-13-2751-2020, 2020.

Lampel, J., Pöhler, D., Tschritter, J., Frieß, U., and Platt, U.: On the relative absorption strengths of water vapour in the blue wavelength range, Atmos. Meas. Tech., 8, 4329–4346, https://doi.org/10.5194/amt-8-4329-2015, 2015.

Wang, H., Souri, A. H., González Abad, G., Liu, X., and Chance, K.: Ozone Monitoring Instrument (OMI) Total Column Water Vapor version 4 validation and applications, Atmos. Meas. Tech., 12, 5183–5199, https://doi.org/10.5194/amt-12-5183-2019, 2019.

---

## Author Comment (AC1) · 1 Jul 2020

We thank reviewer #1 for the useful comments. We understand that these comments are mostly positive while minor corrections are necessary. We have addressed the reviewer's comments on a point to point basis as below for consideration. All page and line numbers refer to the marked-up version of the manuscript.

A new total column water vapor (TCWV) retrieval algorithm in the visible blue spectral band for the Global Ozone Monitoring Experience 2 (GOME-2) instruments on board the EUMETSAT MetOp satellites is presented. The blue spectral band algorithm is important because it allows retrieval of water vapor from sensors which do not cover longer wavelengths, such as Ozone Monitoring Instrument (OMI) and the Coper-

nicus atmospheric composition missions Sentinel-5 Precursor (S5P), Sentinel-4 (S4) andSentinel-5 (S5). This algorithm applies the differential optical absorption spectroscopic (DOAS) technique to retrieve water vapor slant columns and has an iterative optimization module to dynamically find the optimal a priori water vapor profile. This method is better suited for climate studies than usual satellite retrievals with static a priori or vertical profile information from chemistry transport model (CTM). The dynamic a priori algorithm makes use of the fact that the vertical distribution of water vapor is strongly correlated to the total column. The new algorithm is applied to GOME-2A and GOME-2B observations to retrieve TCWV. The validation of this data set includes comparisons to the operational product retrieved in the red spectral band from sunphotometer and radiosonde measurements. These TCWV retrieved in the blue band, which are in good agreement with the other data sets, indicate that the new algorithm derives precise results and can be used for the current and forthcoming Copernicus Sentinel missions S4 and S5.

General comments:

The main motivation for this work is to develop a new water vapor retrieval in the available spectral bands of satellite observations. Further, and this is formulated as requirement also, the retrieval should not rely on input from chemistry transport model (CTM) to avoid propagating model error into the climatological measurement records. Following this, a detailed error estimation is performed. To do so, also water vapor columns in the standard AERONET product as well as temperature, relative humidity, dew point depression, wind direction and wind speed data at multiple pressure levels from radiosondes from stations all over the world are used together with ERA-Interim reanalysis data of water vapor vertical distribution and the relation to their total column amount. The paper addresses relevant scientific questions within the scope of AMT. It completes the knowledge about long-term water vapor content variation of the atmosphere. The paper presents novel concepts, ideas and tools. The scientific methods and assumptions are valid and clearly outlined so that substantial conclusions are

reached. The description of experiments and calculations are sufficiently complete and precise to allow their reproduction by fellow scientists. The quality and information of the figures is fine. The related work is well cited as well as the number and quality of references appropriate i.e. the authors give proper credit to related work and clearly indicate their own new/original contribution. The title and the abstract clearly reflects the contents of the paper. The overall presentation is well structured and clear. The language is fluent and precise. The mathematical formulae, symbols, abbreviations, and units are generally correctly defined.

Specific Comments:

It would be helpful to describe in the beginning of section 2 "Instruments and data sets" to describe the overall evaluation strategy – why these data described here are applied?

Response: We followed the reviewer's comment and supplemented an overall evaluation strategy in the beginning of section 2 (page 4, line 20-23).

Technical corrections:

Not all references include the doi number. It should be unique.

Response: We have carefully gone through the references, some of the references are technical report which doi number is not available. We have supplemented the doi number to the references if it is available. In addition, some of the references include url while some of the references did not. In order to keep it consistence, we have deleted the url for all references.

---

## Author Comment (AC2) · 1 Jul 2020

We thank reviewer #2 for the time to carefully reading the manuscript and providing useful comments. We understand that these comments are positive on the scientific content of the manuscript while appropriate revisions and clarifications are necessary. We have addressed the reviewer's comments on a point to point basis as below for consideration. All page and line numbers refer to the marked-up version of the manuscript.

The paper titled 'Total column water vapor retrieval for GOME-2 visible blue observations' by Chan et al. presents a new method to retrieve total column water vapor(TCWV) from GOME-2 spectra in the blue wavelength band. Slant column water vapor is retrieved using the DOAS spectral fitting technique, and subsequently converted to vertical column using Air Mass Factor (AMF). A dynamic search approach is used to find suitable a priori profile for the AMF calculation. The new algorithm is then applied to retrieve TCWV from GOME-2A and GOME-2B observations. TCWV results from GOME-2A and GOME-2B are also validated against GOME-2 measurements in the red band, sun-photometer and radiosonde measurements. The comparisons show that new blue band retrieval in general shows good agreement with other data sets, and proofed the reliability of the new algorithm. In general, this manuscript presents an interesting results. However, there are still some concerns need to be addressed before publication in AMT.

1) The author introduced the source of a priori water vapor vertical profile in section 3.1.4, which is the statistical result of ERA reanalyzing data for 10 years. In section 2.6, it explained the horizontal and time resolution of ERA data but did not explain the vertical resolution. The vertical resolution and the number of layers should be clearly explained here, because the layer of a priori profile is used to calculate the AMF.

Response: Following the reviewer's suggestion, we have supplemented the number of vertical layer and the resolution of the ERA Interim data set in section 2.6 (page 8, line 3-7).

2) The third line on page 13 of the author indicates that "using a priori profile from model is not optimal for our water vapor retrieval", but the ECMWF ERA data used by the author is also the result of numerical simulation. Here the author only needs to emphasize that the a priori profile used is statistical data, which can reduce the model error.

Response: We followed the reviewer's comment, deleted the sentence and emphasized that the profile information is taken from statistic analysis of historical profiles (page xx, line xx).

3) The author wrote in the third line on page 16 that the profile below cloud is taken from the a priori profile. The water vapor below cloud is close to the atmospheric boundary

layer, where the water vapor concentration is high and the change is large, so this approximation will produce a large error. In addition, should the thickness of the cloud layer also be considered?

Response: The reviewer is right that cloud shield the water vapor column below cloud and hence result in larger error for cloudy scene pixels. The treatment for cloudy pixel follows the independent pixel approximation where the pixel is separated into two independent parts, one with fully cloud cover and the other one is completely cloud free. We have provided a detailed error approximation for cloudy pixel in section 3.1.10. Similar to other tropospheric species, e.g., NO2, we do not recommend user to use data with high cloud fraction. We have also excluded data with intensity weighted cloud fraction larger than 0.5 in our analysis (page 21, line 25). In addition, the thickness of cloud is already considered in the retrieval (see page 16, line 18-21).

4) The authors mentioned the level 1B issue appears at wavelength larger than 460nm, and that is why they avoided including longer wavelength in the spectral analysis. However, we still see a jump of TCWV in 2015 when switching the level 1B data version. The authors should explain why this is still happening even they did not use wavelength longer than 460 nm in their analysis.

Response: The contamination of version 6.0 level 1B data is due to the incomplete removal of Xenon line in the GOME-2 calibration key data. The calibration key data was taken during the preflight on-ground calibration and the calibration key data are used as input for the level 0 to level 1B data processing. The contamination of level 1B data shows negative impact in band 3 while the impact is more significant for wavelength longer 460nm. Therefore, the update of level 1B data also shows an impact on the TCWV product. We have further clarified this point in the manuscript (page 5, line 25-28).

5) The details of the spectral fitting settings are scattered in the text which is quite difficult for the audience to follow. I would suggest the authors to summarize this information in a table.

Response: We have now summarized the spectral fitting details in Table 2.

6) The comparison of TCWV to the retrieval in the red band shows a positive bias over vegetations, e.g., South America and Central Africa, which the authors claim that this is related to the uncertainty of the surface albedo data set. The authors also mentioned that they are trying to improve the albedo database to by using new surface albedo retrieval. It would be nice to show some preliminary results (if available) to show the new method can potentially reduce this bias.

Response: Larger differences between TCWV retrieved in the blue and red band can be observed mainly over vegetation, e.g., south America and central Africa. This is mainly related to the large uncertainty of surface albedo over these areas. We are working on improving the albedo by using a new technique (Loyola et al., 2020). However, this technique has not yet been implemented for GOME-2 measurements in the blue band. Therefore, no preliminary result is available.

7) Page 23, line 15-16: When the authors compare big data sets (sample size > 10k), even a small bias is significant, so I think there is no need and uncommon to mention the P value.

Response: We have deleted the sentence.

---

## Author Comment (AC3) · 1 Jul 2020

We thank reviewer #3 for the useful comments. We have addressed the reviewer's comments on a point to point basis as below for consideration. All page and line numbers refer to the marked-up version of the manuscript.

The manuscript presents a new total column water vapor retrieval algorithm in the visible blue spectral band for the GOME-2 satellite instruments. The blue band algorithm has an advantage over the traditional red band algorithm that allows retrieval of water vapor from sensors which do not cover longer wavelengths, such as OMI and TROPOMI. One of the new features of the algorithm is the dynamic optimization of the a priori water vapor profile, which make use of the fast that the vertical distribution

of water vapor is strongly correlated with the total columns. So the retrieval does not dependent on a priori information from model forecast and better suit for climate study. This paper presented a novel method to retrieve TCWV from satellite observations. Overall, the objective of the study is clear, and the results are validated properly. The presentation of the paper is well structured and easy to follow. Therefore, I suggest publishing the paper after clarifying the minor issues listed below.

1) Please add a summary of the evaluation strategy in section 2 describing why these data are used.

Response: We have supplemented an overall evaluation strategy in the beginning of section 2 (page 4, line 20-24).

2) The Level 1 data issue is still not clear to me, usually satellite measurement quality is decaying over time, why the data after 2015 is fine but the data before 2015 is contaminated? Is there any recalibration done in 2015? If the recalibration improves the data quality, will there be any reprocessing?

Response: This is related to the switch of level 1B processor in 2015. The level 1B processor has been update from version to 6.0 to 6.1 in 2015. This update mainly improved the spectral artefacts in the GOME-2 on-ground calibration key data. The spectral artefact in the level 1B data is due to incomplete removal of Xenon line in the GOME-2 calibration key data. The calibration key data was taken during the preflight on-ground calibration and the calibration key data are used as input for the level 0 to level 1B data processing. Therefore, the update of the processor improved the level 1B data and subsequently improved the TCWV data. We have further clarified this in the manuscript (page 5, line 25-28). The level 1B data is processed at EUMETSAT, and we cannot tell if there will be any reprocessing in the future.

3) The error due to surface albedo listed in table 3 is 3%, while the bias over vegetation, e.g., Amazon and Central Africa, seems to be much larger. Please clarify.

Response: The value listed in the Table is for typical cases, while the uncertainty of surface albedo over vegetation, e.g., Amazon and Central Africa, are larger, and therefore, resulting in larger uncertainty over these areas. We have further clarified in the manuscript that the values list on the Table are typical values while exceptional cases might exceed these values (page 20, line 16-17).

4) Why the absorption cross section of NO2 is not âĹij290k

Response: It is because most of the NO2 is in the stratosphere where the temperature is much lower than 220k. Therefore, NO2 cross section with lower temperature is used in the retrieval. NO2 Cross section measured with lower temperature is typically used also for the retrieval of NO2 columns, e.g., Liu et al., 2019.

Liu, S., Valks, P., Pinardi, G., De Smedt, I., Yu, H., Beirle, S., and Richter, A.: An improved total and tropospheric NO2 column retrieval for GOME-2, Atmospheric Measurement Techniques, 12, 1029–1057, https://doi.org/10.5194/amt-12-1029-2019, 2019.

5) Some of the references are inconsistence in format.

Response: We have carefully gone through the references, some of the references are technical report which doi number is not available. We have supplemented the doi number to the references if it is available. In addition, some of the references include url while some of the references did not. In order to keep it consistence, we have deleted the url for all references.
* * *

---

## Author Comment (AC4) · 1 Jul 2020

We thank for the constructive comments. Our response to the comments is below. All page and line numbers refer to the marked-up version of the manuscript.

Chan et al. present a very interesting TCWV retrieval for GOME-2 which not only makes use of the H2O absorption in the blue spectral range, but also applies an iterative a priori water vapour profile approach. The authors are probably aware that Borgeret al. (2020) developed a very similar approach. I have a few comments/questions that can hopefully help to further improve the overall high quality of the paper:

Response: We have to admit that the literature cited in the manuscript might not be up to date. Publication published in the past half a year was not included, as the algorithm

was developed while ago. We have now included your publication in the reference.

1. One great advantage of using the blue spectral range compared to the red spectral range is the higher sensitivity for the near-surface layers over ocean. Did the authors also consider to compare their TCWV data set to microwave satellite sensors? For instance SSMI/SSMIS are widely considered as reference measurements for TCWV retrievals over ocean because they can measure under all-sky conditions. It would be very important to see how good the TCWV from GOME-2 can match the TCWV from SSMI/SSMIS.

Response: We agree with you that SSMI/SSMIS provides measurements over ocean where the GOME-2 measurements in the blue band show higher sensitivity. On the other hand, the GOME-2 measurements in the red band have been validated against SSMI/SSMIS data and the two data sets show very good agreement (Grossi et al., 2015). The comparison between GOME-2 blue and red band retrieval also show excellent agreement. Therefore, we can expect that the new data set matches the SSMI/SSMIS observations. In addition, we have compared the new product to sun-photometer and radiosonde measurements and both results are consistence. As the focus of the manuscript is the algorithm development, we cannot handle everything within one manuscript, so we would like to leave the comparison to SSMI/SSMIS data for the follow up validation study.

2. A similar question is related to the use of AERONET data as reference. As the authors mention it themselves, TCWV from AERONET is potentially affected by biases. Have the authors had a look at TCWV from ground-based GPS measurements from SuomiNet or IGS? In comparison to AERONET, these GPS networks can conduct TCWV retrievals for all-sky conditions at a very high accuracy and provide continuous time series of TCWV for their measurement stations (e.g. SuomiNet TCWV data are available every 30 min). Such a comparison would further improve the confidence of the new retrieval.

Response: Thanks for the suggestion. All measurements have their own advantages and disadvantages. The uncertainty of GPS data is also strongly dependent on temperature and pressure profile. For sure, the new product may benefit from comparing to GPS data. However, we have already compared the GOME-2 TCWV product to three different measurements and we cannot take care everything within the manuscript. Therefore, we would like to leave the comparison for the next validation study.

3. It is kind of surprising to see that the GOME-2 TCWV retrieval uses HITEMP2010 /HITRAN2012 as $H_2O$ cross-section, whereas Wang et al. (2019) (for OMI) and Borgeret al. (2020) (for TROPOMI) found a significantly better agreement to reference measurements by using HITRAN2008. The use of HITRAN2008 over HITRAN2012 is also supported by the LP-DOAS results in Lampel et al. (2015) (see Table 8 in their paper). What is the rationale for still using HITRAN2012?

Response: We have carefully checked the available cross sections while developing the retrieval algorithm. The water vapor cross section from the HITRAN 2008 data set is also included in the sensitivity study. Water vapor SCDs retrieved with the HITRAN 2008 cross section are 1-2% higher than the one retrieved with the corrected cross section in HITEMP2010. The enhancement of SCDs is more significant over higher altitude and would further enhances the positive bias over these areas. In addition, the root mean square of the spectral fit residual using the HITEMP 2010 cross section is slightly ($\sim$3%) smaller. Therefore, we decided to use the HITEMP 2010 cross section instead of the other one. We have provided this information in the manuscript (page 11, line 1-6).

4. Although it is very reasonable to use ERA-Interim for the statistical analysis, isn't there potentially the risk that the data quality of the ERA-Interim water vapour profile scan vary a lot e.g. depending on which measurement data have been used during the data assimilation process? At least this was one of the reasons for Borger et al. (2020) to only use water vapour profiles from a consistent measurement data set (COSMIC in this case) for setting up their iterative a priori water vapour profile retrieval scheme.

Response: The ERA Interim data assimilated multiple sources of data including radiosonde and many ground and satellite based remote sensing observations, which could potentially avoid bias from single source. In addition, we are using statistical data over a long period (11 years) the bias or uncertainty caused by different observations is expected to be negligible compared to other sources of error. A more comprehensive comparison of different data assimilation results would provide a more solid conclusion on this topic, however, it is beyond the scope of this manuscript.

References

Borger, C., Beirle, S., Dörner, S., Sihler, H., and Wagner, T.: Total column water vapour retrieval from S-5P/TROPOMI in the visible blue spectral range, Atmos. Meas. Tech.,13, 2751–2783, https://doi.org/10.5194/amt-13-2751-2020, 2020.

Lampel, J., Pöhler, D., Tschritter, J., Frieß, U., and Platt, U.: On the relative absorption strengths of water vapour in the blue wavelength range, Atmos. Meas. Tech., 8, 4329–4346, https://doi.org/10.5194/amt-8-4329-2015, 2015.

Wang, H., Souri, A. H., González Abad, G., Liu, X., and Chance, K.: Ozone Monitoring Instrument (OMI) Total Column Water Vapor version 4 validation and applications, Atmos. Meas. Tech., 12, 5183–5199, https://doi.org/10.5194/amt-12-5183-2019, 2019.